# Technical note: Effects of Uncertainties and Number of Data points on Line Fitting - a Case Study on New Particle Formation

Santtu Mikkonen[1], Mikko R. A. Pitkänen[1,2], Tuomo Nieminen[1§], Antti Lipponen[2], Sini Isokääntä[1], Antti Arola[2], and Kari E. J. Lehtinen[1,2]

[1] Department of Applied Physics, University of Eastern Finland, Kuopio, Finland
[2] Finnish Meteorological Institute, Atmospheric Research Centre of Eastern Finland, Kuopio, Finland
[§] Currently at: Institute for Atmospheric and Earth System Research, University of Helsinki, Helsinki, Finland

*Correspondence to*: Santtu Mikkonen (santtu.mikkonen@uef.fi)

**Abstract.** Fitting a line of two measured variables is considered one of the simplest statistical procedures researchers can do. However, this simplicity is deceptive as the line fitting procedure is actually quite a complex problem. Atmospheric measurement data never comes without some measurement error. Too often, these errors are neglected when researchers are making inferences from their data.

To demonstrate the problem, we simulated datasets with different amounts of data and error, mimicking the dependence of atmospheric new particle formation rate ($J_{1.7}$) on sulphuric acid concentration ($H_2SO_4$). Both variables have substantial measurement error and thus they are good test variables for our study. We show that ordinary least squares (OLS) regression results in strongly biased slope values compared with six error-in-variables (EIV) regression methods (Deming, Principal component analysis, orthogonal, Bayesian EIV, and two different bivariate regression methods) known to take into account errors in the variables.

## 1  Introduction

Atmospheric measurements always come with some measurement error. Too often, these errors are neglected when researchers are making inferences based on their data. Describing the relationship between two variables typically involves making deductions in some more general context than was directly studied. If the relationship is not defined correctly, the inference is not valid either. In some cases, the bias in analysis method is even given a physical meaning.

When analysing dependencies of two or more measured variables, regression models are usually applied. Regression models can be linear or non-linear, depending on the relationship between data sets that are analysed. . Standard regression models assume that the independent variables of the model have been measured without error and the model account only for errors in the dependent variables or responses. In cases where the measurements of the predictors contain error, estimating with standard methods, usually Ordinary Least Squares (OLS), do not tend to the true parameter values, not even with very high number of number of data points. In linear models, the coefficients are underestimated (e.g. Carroll et al., 2006) but in nonlinear

models, the bias is likely to be more complicated (e.g. Schennach 2004). If predictor variables in regression analysis contain any measurement error, methods that account for errors should be applied. Particularly when errors are large. Thus, test variables in this study were chosen such that they included significant uncertainties in both the independent and dependent variables. Sulphuric acid ($H_2SO_4$) is known to strongly affect the formation rates ($J$) of aerosol particles (Kirkby et al., 2016;

Kuang et al., 2008; Kulmala et al., 2006; Kürten et al., 2016; Metzger et al., 2010; Riccobono et al., 2014; Riipinen et al., 2007; Sihto et al., 2006; Spracklen et al., 2006). The relationship between $J$ ($cm^{-3}$ $s^{-1}$) and $H_2SO_4$ (molec $cm^{-3}$) is typically assumed to be in form $\log_{10}(J) = \beta*\log_{10}(H_2SO_4)+\alpha$ (Seinfeld and Pandis, 2016). In addition, parameterizations based on the results from these fits have been implemented in global models, e.g. in (Dunne et al., 2016; Metzger et al., 2010; Spracklen et al., 2006), to estimate the effects of new particle formation on global aerosol amounts and characteristics. Theoretically in

homogeneous nucleation, the slope of this relationship is related to the number of sulphuric acid molecules in the nucleating critical cluster, based on the first nucleation theorem (Vehkamäki, 2006)..

Some published results have shown discrepancies in the expected $J$ vs $H_2SO_4$ dependence. Analysing data from Hyytiälä in 2003, Kuang et al. (2008) used an unconstrained least squares method, which was not specified in the paper, and obtained $\beta$=1.99 for the slope, whereas Sihto et al. (2006) reported a value of $\beta$=1.16 using OLS from the same field campaign. They

had some differences in pre-treatment of data and used different time windows, but a significant proportion of this inconsistency is very likely due to use of different fitting methods. The problem in the relationship of $H_2SO_4$ and $J$ has been acknowledged previously in Paasonen et al. (2010) who noted that bivariate fitting method as presented in York et al. (2004) should be applied but could not be used due to the lack of proper error estimates for each quantity. They were not aware of the methods that do not need to know the errors in advance, but instead made use of estimated variances. Here, we present

appropriate tools for using that approach.

Multiple attempts have been made to present methods accounting for errors in predictor variables for regression-type analysis, going back to Deming (1943). However, the traditional least squares fitting still holds the position as the de facto line fitting method due to its simplicity and common availability in frequently used software. In atmospheric sciences, Cantrell (2008) drew attention to the method introduced by York (1966) and York et al. (2004) and listed multiple other methodological papers

utilizing similar methodology. Pitkänen et al. (2016) raised the awareness of the problem in remote sensing community and this study partly follows their approach and introduces multiple methods to take account the errors in predictors. Cheng and Riu (2006) studied methods with heteroscedastic errors whereas Wu and Yu (2018) approached the problem with measurement errors via weighted regression and applied some techniques also used in our study.

Measurement errors in each variable must be taken into account using approaches called errors-in-variables (EIV) regression.

EIV methods simply mean that errors in both variables are accounted for. In this study, we compared OLS regression results to six different regression methods (Deming regression, Principal component analysis regression, orthogonal regression, Bayesian EIV regression and two different bivariate regression methods) known to be able to take into account errors in variables and provide (at least asymptotically) unbiased estimates. In this study, we will focus only on linear EIV methods but it is important to acknowledge that there also exist nonlinear methods e.g. ORDPACK introduced in Boggs, Byrd, and Schnabel

(1987) and implemented in Python SciPy and R (Boggs et al., 1989; Spiess, 2015). ORDPACK is a somewhat improved version of classical orthogonal regression, so that arbitrary covariance structures are acceptable and is specifically set up so that a user can specify measurement error variances and covariance point by point, as some of the methods in this study are doing in linear analysis.

## 2    Materials and Methods

### 2.1    Data illustrating the phenomenon

Measurement data contains different types of errors. Usually, the errors are divided to two main class: random and systematic error. Systematic errors, commonly referred as bias, in experimental observations usually come from the measuring
instruments. They may occur because there is something wrong with the instrument or its data handling system, or because the instrument is not used correctly by the operator. In line fitting, bias cannot be taken account but it needs to be minimized through careful and regular instrument calibrations and zeros or data pre-processing. The random error instead may have different components, of which two are discussed here: natural error and measurement error. In addition, one should note the existence of equation error, discussed in Carroll and Ruppert (1996), which refers to using an inappropriate form of a fitting
equation. Measurement error is more generally understood, it is where measured values do not fully represent the true values of the variable being measured. This also contains sampling error, e.g. in the case of $H_2SO_4$ measurement the sampled air in the measurement instrument is not representative sample of outside air (e.g. due to losses of $H_2SO_4$ occurring in the sampling lines). Natural error is the variability caused by natural or physical phenomenon e.g. certain amount of $H_2SO_4$ does not cause same number of new particles formed. In the analysis of measurement data, some amount of these errors are known or can be
estimated, but some of it will usually remain unknown, which should be kept in mind when interpreting fits. Even though the measurement error is taken into account, the regression fit may be biased due to unknown natural error. In this study, we assume that the errors of the different variables are uncorrelated, but in some cases this has to be taken into account, as noted e.g. in Trefall and Nordö (1959) and Mandel (1984). The correlation between the errors of two variables, measured with separate instruments, independent of each other, like formation rate and $H_2SO_4$, may come e.g. from environmental variables
affecting both of them at the same time. Factors affecting formation of sulphuric acid have been studied in various papers, e.g. in Weber et al. (1997) and Mikkonen et al. (2011). New particle formation rates, in turn, have been studied e.g. in Boy et al.( 2008) and in Hamed et al. (2011) and similarities between affecting factors can be seen. In addition, factors like room temperature in the measurement space and atmospheric pressure may affect the performance of instrumentation, thus causing additional error.
The data used in this study consist of simulated new particle formation rates at 1.7 nanometre size  ($J_{1.7}$) and sulphuric acid ($H_2SO_4$) concentrations mimicking observations of pure sulphuric acid in nucleation experiments from the CLOUD chamber

in CERN (Kürten et al. 2016; https://home.cern/about/experiments/cloud) with corresponding expected values, their variances and covariance structures. The Proton Synchrotron provides an artificial source of "cosmic rays" that simulates natural conditions of ionization between ground level and the stratosphere. The core is a large (volume 26m3) electro-polished stainless steel chamber with temperature control (temperature stability better than 0.1 K) at any tropospheric temperature,

precise delivery of selected gases ($SO_2$, $O_3$, $NH_3$, various organic compounds) and ultrapure humidified synthetic air, and very low gas-phase contaminant levels. The existing data on NPF includes what are believed to be the most important routes that involve sulphuric acid, ammonia and water vapour (Kirkby et al., 2011), sulphuric acid – amine (Almeida et al., 2013) and ion induced organic nucleation (Kirkby et al., 2016).The actual nucleation of new particles occurs at slightly smaller size. After formation, they grow by condensation to reach the detection limit (1.7 nm) of the instrument and $J_{1.7}$ thus refers to the formation

rate of particles as the instrument detects them, taking into account the known particle losses due to coagulation and deposition on the chamber walls. The relationships between precursor gas phase concentrations and particle formation rates were chosen because they are both known to have considerable measurement errors and their relationship is studied frequently using regression-based analyses (Kirkby et al., 2016; Kürten et al., 2016; Riccobono et al., 2014; Tröstl et al., 2016). Additionally, many of the published papers on this topic do not describe how they are taking account the uncertainties in the analysis, which

casts doubt that errors have been treated properly. However, it should be kept in mind that the data could be any set of numbers assumed to have linear relationship, but in order to raise awareness in the aerosol research community, in this study we relate our analysis to the important problem of understanding new particle formation.

## 2.2    Regression methods

We made fits for the linear dependency of logarithms of the two study variables, such that the equation for the fit was given by

$$y = \beta_0 + \beta_1 x + \varepsilon \tag{1}$$

where $y$ represents $\log_{10}(J_{1.7})$, $x$ is $\log_{10}(H_2SO_4)$, $\beta$'s are the coefficients estimated from the data and $\varepsilon$ is the error term. In order to demonstrate the importance of taking into account the measurement errors in the regression analysis, we tested seven

different line-fitting methods. Ordinary Least Squares (OLS), not taking account the uncertainty in $x$-variable, and orthogonal regression (ODR, Boggs, Byrd, and Schnabel 1987), Deming regression (DR, Deming, 1943), Principal component analysis (PCA, Hotelling, 1957) regression, Bayesian EIV regression (Kaipio and Somersalo, 2005) and two different bivariate least squares methods by York *et al.*, (2004), and Francq and Govaerts (BLS, 2014), known to be able to take account errors in variables and provide (at least asymptotically) unbiased estimates. The differences between the methods come from the

criterion they minimize when calculating the coefficients and how they account for measurement errors. The minimizing criteria for all methods are given in Appendix A1, but here we give the principles of the methods. OLS minimizes the squared distance of the observation and the fit line either in y or x direction, but not both at the same time, whereas ODR minimizes the sum of squared weighted orthogonal distances between each point and the line. DR was originally an improved version of

orthogonal regression, taking account the ratio of the error variances, $\lambda_{xy}$, of the variables, (in classical non-weighted ODR $\lambda_{xy}$ =1) and it is the maximum likelihood estimate (MLE) for the model (1) when $\lambda_{xy}$ is known. The approach of PCA is the same as in ODR but the estimation procedure is somewhat different as can be seen in S1. The bivariate algorithm by York et al 2004 provides a simple set of equations for iterating MLE of slope and intercept with weighted variables, which makes it similar to

ODR in this case. However, using ODR allows for performing regression on a user defined model, while the York (2004) solution works only on linear models. This, for instance, enables using linear scale uncertainties in ODR in this study, while the York (2004) approach could only use log scale uncertainties. In Bayes EIV, statistical models for the uncertainties in observed quantities are used and probability distributions for the line slope and intercept are computed according to the Bayes' theorem. In this study, we computed the Bayesian maximum a posteriori (MAP) estimates for the slope and intercept that are

the most probable values given the likelihood and prior models, see Appendix A1 for more details on models used in Bayes EIV. BLS takes into account errors and heteroscedasticity, i.e. unequal variances, in both variables and thus is more advanced method than DR (under normality and equal variances, BLS is exactly equivalent to DR). PCA accounts only for the observed variance in data, whereas ODR, Bayes EIV and York bivariate regression require known estimates for measurement errors. Though for Bayes EIV the error can be approximated with a distribution. DR and BLS can be applied with both errors given

by the user and measurement variance based errors. In this study, we applied measurement variance based errors for them. The analysis for OLS and PCA were calculated with R-functions "lm" and "prcomp", respectively (R Core Team, 2018) DR was calculated with package "deming" (Therneau, 2018) and BLS with package "BivRegBLS" (Francq and Berger, 2017) in R. The ODR based estimates were obtained using "scipy.odr" python package (Jones et al., 2001), while the python package "pystan" (Stan Development Team, 2018) was used for calculating the Bayesian regression estimates. Finally, the York

bivariate estimates were produced with a custom python implementation of the algorithm presented by York et al. (2004).

## 3    Data

### 3.1    Simulated data

In measured data, the variables that are observed are not $x$ and $y$, but $(x+e_x)$ and $(y+e_y)$, where $e_x$ and $e_y$ are the uncertainty in the measurements, and the true $x$ and $y$ cannot be exactly known. Thus, we used simulated data, where we know the true, i.e.

noise-free $x$ and $y$, to illustrate how the different line fitting methods perform in different situations.

We simulated a dataset mimicking new particle formation rates ($J_{1.7}$) and sulphuric acid concentrations ($H_2SO_4$) reported from CLOUD-chamber measurements in CERN. Both variables are known to have substantial measurement error and thus they are good test variables for our study. Additionally, the relationship of logarithms of these variables is quite often described with linear OLS regression and thus the inference may be flawed.

We generated one thousand random noise-free $H_2SO_4$ concentration values assuming log-normal distribution with median $2.0*10^6$ *(molecules cm⁻³)* and standard deviation $2.4*10^6$ *(molecules cm⁻³)*. The corresponding noise-free $J_{1.7}$ was calculated

using model $\log_{10}(J_{1.7}) = \beta*\log_{10}(\underline{H_2SO_4})+\alpha$ with the noise-free slope $\beta$=3.3 and $\alpha$=-23, both are realistic values presented by Kürten *et al.* (2016, Table 2 for the no added ammonia cases). (Kürten et al., 2016)

Simulated observations of the noise-free $H_2SO_4$ were obtained by adding random errors $e_x = e_{rel,x}x + \sigma_{abs,x}$ that have a random absolute component $e_{abs,x} \sim normal(0,\sigma_{abs,x})$ and a random component relative to the observation $x$ itself $e_{rel,x}x$, where $e_{rel,x} \sim normal(0,\sigma_{rel,x})$. Similar definitions apply for the noise-free $J_{1.7}$, $e_y$, $\sigma_{abs,y}$ and $\sigma_{rel,y}$. The standard deviations of the measurement error components were chosen $\sigma_{abs,x} = 4*10^5$, $\sigma_{rel,x} = 0.3$, $\sigma_{abs,y} = 3*10^{-3}$, $\sigma_{rel,y} = 0.5$, which are subjective estimates based on measurement data. The resulting total errors were occasionally about as large as the data values themselves, but they are not unusually large error values in corresponding real datasets, where overall uncertainties may reach 150 % for H2SO4 concentrations and 250 % for nucleation rates (e.g. Dunne et al., 2016).

These choices of generating simulated data reflect what real data set can often be like: the bulk of the data approximates a log-normal distribution with possibly one of the tails thinned or cut close to a limit of detection of an instrument or close to a limit of data filtering criterion. In our simulated data, each negative observation and each negative noise-free value was replaced with a new random simulated value, which only slightly offsets the final distribution from a perfectly symmetric log-normal shape.

Simulating the observations tends to generate infrequent extreme outlier observations from the infinite tails of the normal distribution. We discarded these outliers with absolute error larger than three times the combined standard uncertainty of the observation in order to remove the effect of outliers from the regression analysis. This represents the quality control procedure in data analysis and it also improved the stability of our results between different simulations.

## 3.2    Case study on measured data

In order to show that the results gained with simulated data are applicable also in real measurement data, we applied our methods to data measured in CLOUD chamber and published by Dunne et al. (2016). Dunne et al. Fig 1. shows nucleation rates (*J*) at 1.7 nm mobility diameter as a function of sulphuric acid concentration. We used their measurements with no added ammonia in two different temperatures, 278K and 292K, shown in their Fig 1 subplots (D) and (E) and published as supplemental data.

# 4    Results

## 4.1    Fits for simulated data

Differences between the regression methods are illustrated with four different ways. Firstly, by showing line fits on scatterplot of simulated data. Secondly, illustrating how the slopes change when the uncertainty in the measured variables increase, thirdly by showing the sensitivity of the fits on number of observations and finally showing how the fits are affected by adding outliers in the data.

Regression fits with all methods in use are shown in Figure 1. As we know that the noise-free slope $\beta_{true}$=3.30 we can easily see how the methods perform. The worst performing method was OLS, with $\beta_{ols}$=1.55, which is roughly half of the $\beta_{true}$. The best performing methods with equal accuracy, i.e. within 2% range, were ODR ($\beta_{ODR}$=3.27), Bayes EIV ($\beta_{BEIV}$=3.24) and BLS ($\beta_{BLS}$=3.22), whereas York ($\beta_{York}$=3.15) was within 5% range, but Deming ($\beta_{DR}$=2.95) and PCA ($\beta_{PCA}$=2.92) slightly

underestimated the slope.

The sensitivity of the methods was first tested by varying the uncertainty in $H_2SO_4$ observations. We simulated six datasets with 1000 observations and with varying absolute and relative uncertainties, listed in Table 1, and performed fits with each method on all of these datasets. The performance of the methods is shown in Figure 2, with the results corresponding to Figure 1 marked in black. It shows that when the uncertainty is small, the bias in OLS fit is smaller but when more uncertainty is

added to data the bias increases significantly. Decrease in performance can also be seen with ODR, which overestimates the slope, and PCA, DR and Bayes EIV, which all underestimate the slope. Bivariate methods, BLS and York, seem to be quite robust with increasing uncertainty, as the slopes are not changing significantly.

The sensitivity of methods to decreasing number of observations was tested by picking 100 random samples from the 1000 simulation dataset with $n$ of 3, 5, 10, 20, 30, 50, 70, 100, 300 and 500 and making fits for all samples with all methods. The

average slopes and their standard errors are shown in Figure 3. It is clear that when the number of observations is 10 or less, the variation in estimated slopes can be considerably high. When $n \geq 30$ the average slopes stabilized close to their characteristic levels (within 5%), except for Bayes EIV and York bivariate, which needed more than 100 observations. The most sensitive methods for small $n$ were Bayes EIV, ODR and PCA and thus they should not be applied for data with small $n$ and similar type of uncertainty than presented here. Though, it should be remembered that number of points needed for a good fit depends

on the uncertainties in the data.

The sensitivity for outliers in predictor variable $H_2SO_4$ was tested with two different scenarios. First, the outliers were let to be randomly either high or low end of the distribution. In the second scenario, outliers were allowed to be only large numbers, which is often the case in $H_2SO_4$ and aerosol concentration measurements as the smallest numbers are cleaned out from the data when they are smaller than the detection limit of the measurement instrument. Five cases with n=1000 were simulated

with increasing number of outliers (0, 5, 10, 20, 100) and 10 repetitions of $H_2SO_4$ values with different set of outliers. Outliers were defined such that $x_{obs}-x_{true}$>3*combined standard uncertainty. The most sensitive methods for outliers in both scenarios were OLS and Bayes EIV. High number of outliers caused underestimation to PCA and DR, especially in high outlier case, and slight overestimation to BLS in random outlier case. York Bivariate and ODR were not affected in either case and BLS had only small variation between the 10 replicates in the estimated slope. We did not explore how large a number of outliers

would be needed to seriously disrupt the fits for the various methods. We felt that it is likely not realistic to have situations with more than 10% outliers.

We also applied an alternative method for simulating the data for testing different methods. The main difference compared to our method was, that the distribution of noise-free $H_2SO_4$ followed uniform distribution in log-space. With this assumption, it could be seen that OLS works almost equally well compared to EIV methods introduced here if the range of data is wide

($H_2SO_4$ concentration in range $10^6$-$10^9$). However, when scaled to concentrations usually measured in the atmosphere ($10^4$-$10^7$) the high uncertainties caused similar behaviour to data than seen in our previous simulations. Details of these results can be seen in Supplement S1.

## 5    4.2    Results on the case study

Figure 5 shows the fits on the data from Dunne et al (2016). As expected, the fit with OLS is underestimated with both temperatures ($\beta_{ols}$(278K) = 2.4 and $\beta_{ols}$(292K) = 3.0). The regression equations for all methods are shown in Figure 5. Dunne et al. did not use linear fit in their study but applied nonlinear Levenberg-Marquardt algorithm (Moré, 1978) instead on function $J_{1.7} = k*[H_2SO_4]^\beta$ where $k$ is temperature dependent rate coefficient with nonlinear function including three estimable
parameters (see section 8 in their Supplement for details). Thus, the results are not directly comparable as, for simplicity, we made the fits to data measured in different temperatures separately. However, their $\beta$-value for the fit ($\beta$=3.95) is quite close to our results with EIV-methods, especially slopes from Bayes EIV in 292K and BLS and PCA in both temperatures were within 5% range. We also made some tests for data measured in lower temperatures, results not shown here. But as a summary, the slopes did not vary drastically from those in $\beta_{ols}$(278K) and $\beta_{ols}$(292K) when the other conditions were similar, even though
the lower number of observations in lower temperatures increased uncertainty in the data. However, the intercepts $\beta_0(T)$ varied between temperatures.

## 5    Conclusions

Ordinary least squares regression can be used to answer some simple questions on data, such as "How is $Y$ related to $X$?". However, if we are interested in the strength of the relationship and the predictor variable $X$ contains some error, then error-
in-variables methods should be applied. There is no single correct method to make the fit, because the methods behave slightly differently with different types of error. The choice of method should be based on the properties of data and the specific research question. There are usually two types of error in the data: natural and measurement error, where natural error refers to stochastic variation in the environment. Even if the natural error in the data is not known, taking into account the measurement error improves the fit significantly. Weighting the data based on some factor, typically the inverse of the
uncertainty, reduces the effect of outliers and makes the regression depend more on the data that is more certain (see e.g. Wu and Yu, 2018) but it does not solve the problem completely.

As a test study, we simulated a dataset mimicking the dependence of atmospheric new particle formation rate on sulphuric acid concentration. We introduced three major sources of uncertainty when doing inference from scatterplot data: increasing measurement error, number of data points and number of outliers. In Fig 1, we showed that in case of simulations where errors
are taken from real measurements of $J_{1.7}$ and $H_2SO_4$ four of the methods gave slopes within 5% of the known noise-free value: BLS, York bivariate, Bayes EIV and ODR. Estimates from BLS and York bivariate remained stable even when the uncertainty

in simulated $H_2SO_4$ was increased drastically in Fig 2. The main message to learn in Fig 3 is that if the data contain some error, then with small numbers of observations all fit methods are highly uncertain. BLS was the most accurate with smallest sample sizes of 10 and less, ODR stabilized with 20 observations and York bivariate and Bayes EIV needed 100 or more data points to become accurate. After that, they approach the noise-free value asymptotically, while the OLS slope, in contrast, converges towards an incorrect value. With the increasing number of outliers (Figure 4) ODR and York bivariate were the most stable ones, even with 10% of observations classified as outliers in both test cases. BLS remained stable in the case with only high outliers. Bayes EIV was the most sensitive to outliers after OLS.

From this, we can give a recommendation that if the uncertainty in predictor is known, York bivariate, or other method able to use known variances, should be applied. If the errors are not known, and they are estimated from data, BLS and ODR showed out to be the most robust in cases of increasing uncertainty (relative error rE > 30% in Fig 2) and with high number of outliers. In our test data, BLS and ODR stayed stable up to rE >80% in Fig. 2 whereas DR and PCA started to be more uncertain when rE > 30% and Bayes EIV when rE>50%. If the number of observations is less than 10, and the uncertainties are high, we recommend considering if a regression fit is appropriate at all. However, with chosen uncertainties in our simulation tests BLS showed out to be the most robust with small numbers of data points. Bayes EIV has significant advantages if the number of observations is high enough and there are not too many outliers, as it does not require explicit definition of the errors but can treat them as unknown parameters given their probability distributions.

We also made a case study on data measured in CLOUD chamber and published by Dunne et al. (2016). In these analyses, we saw that our recommended methods above are performing best also for these data. Our tests indicated that the slope $\beta_1$ for the fit is not highly sensitive for changes in temperature in the chamber but the intercept $\beta_0$ in linear fit is. This dependency was also seen, and taken account, in Dunne et al. (2006).

**Author contribution**

SM prepared the manuscript with contributions from all co-authors. SM, MP and SI performed the formal analysis. MP simulated the data. SM, AA and KL formulated the original idea. SM, MP and AL developed and implemented the methodology. SM, MP, TN and AL were responsible for investigation and validation of data and methods.

**Acknowledgments**

This work was supported by The Nessling foundation and The Academy of Finland Centre of Excellence (grant no. 307331).

**Competing interests**

The authors declare that they have no conflict of interest.

*Code availability:* Python code for running the methods can be found in GitHub:
https://gist.github.com/mikkopitkanen/da8c949571225e9c7093665c9803726e

*Data availability*: Simulated datasets used in the example analysis are given as supplement.

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

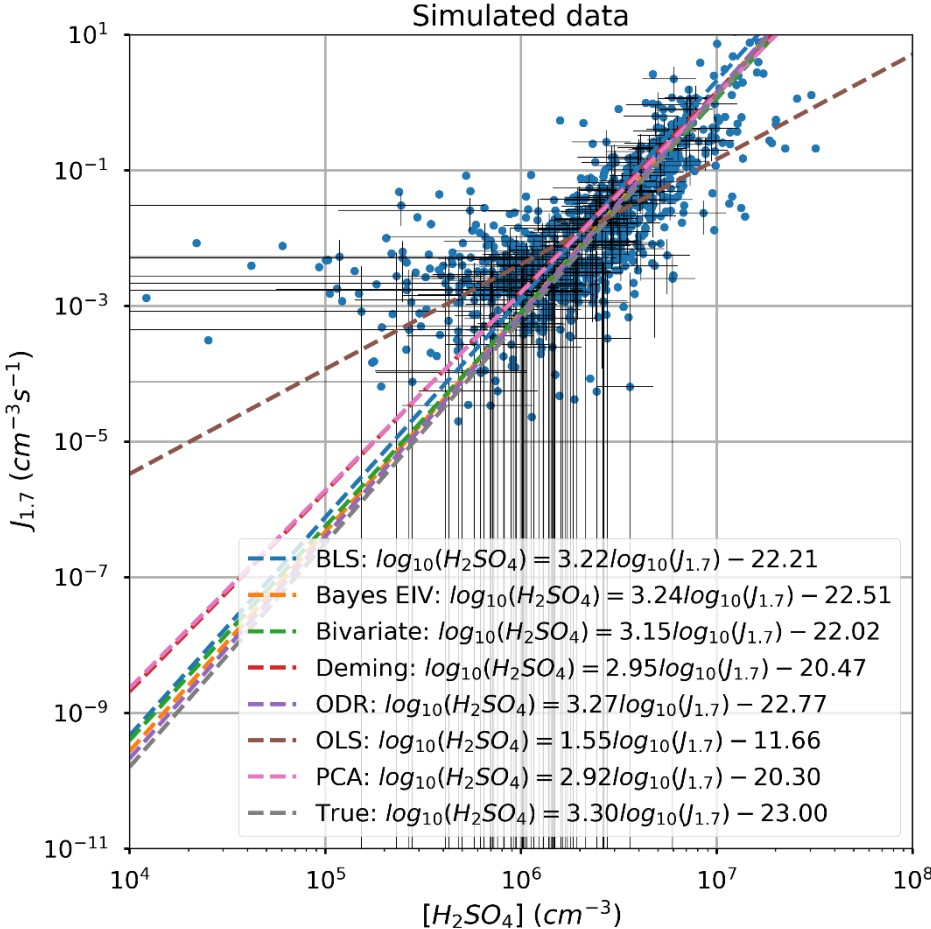

**Figure 1. Regression lines fitted to the simulated data with all methods in comparison. Whiskers in data points refer to the measurement error used for simulation**

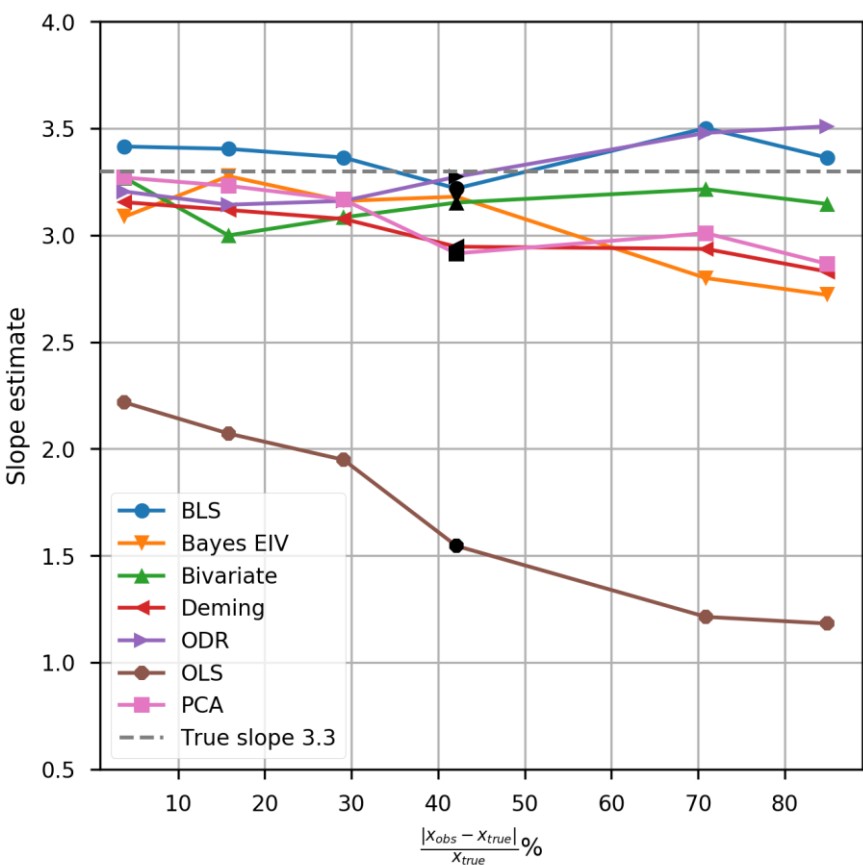

**Figure 2. Sensitivity test for increasing uncertainty in simulated data. Black markers show the initial data set described in Section 3. Dashed line indicates the noise-free slope.**

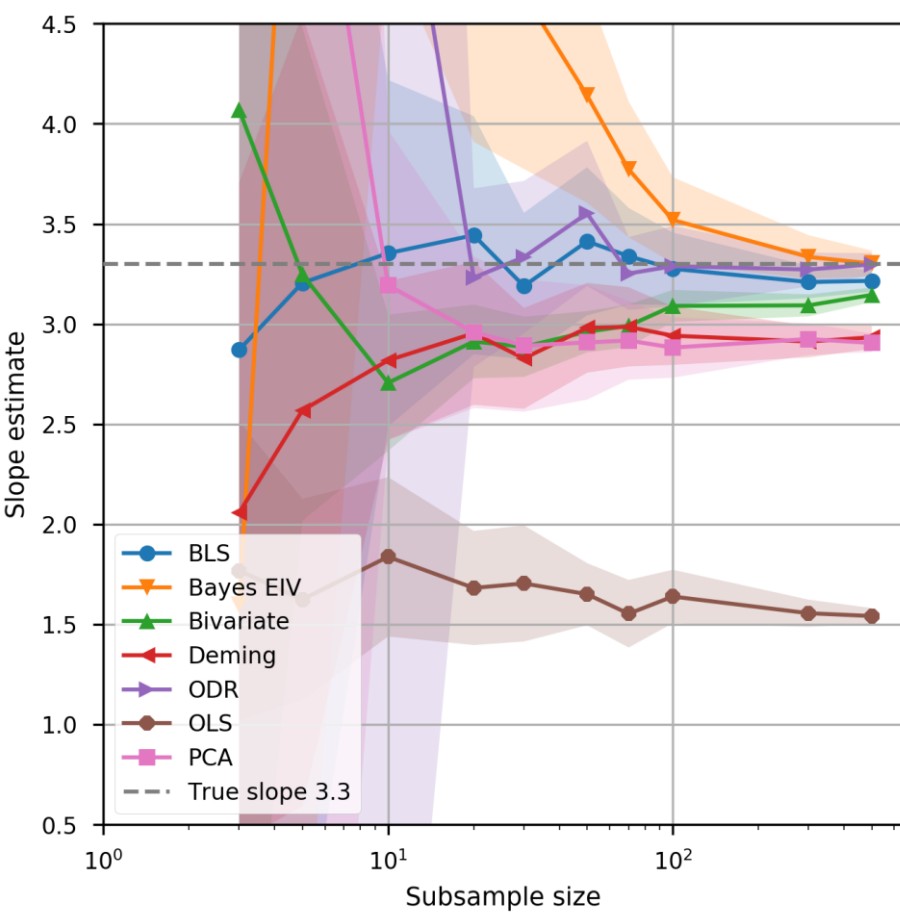

**Figure 3. Effect of sample size on the uncertainty of different fits. Lines show the median and shading illustrates one standard deviation range of slope estimates for 40 repeated random samples. Dashed line indicates the noise-free slope.**

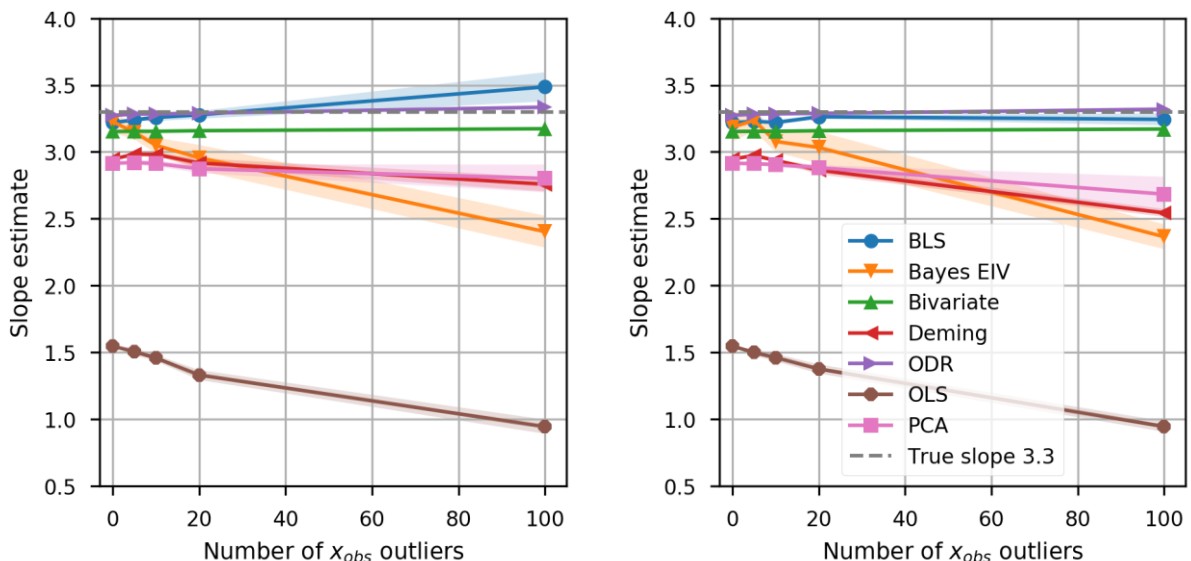

**Figure 4. Effect of outliers in the data. Random outliers case on left panel and only high positives on right panel. Lines show the median and shading shows one standard deviation of slope estimates in ten repeated studies. Dashed line indicates the noise-free slope.**

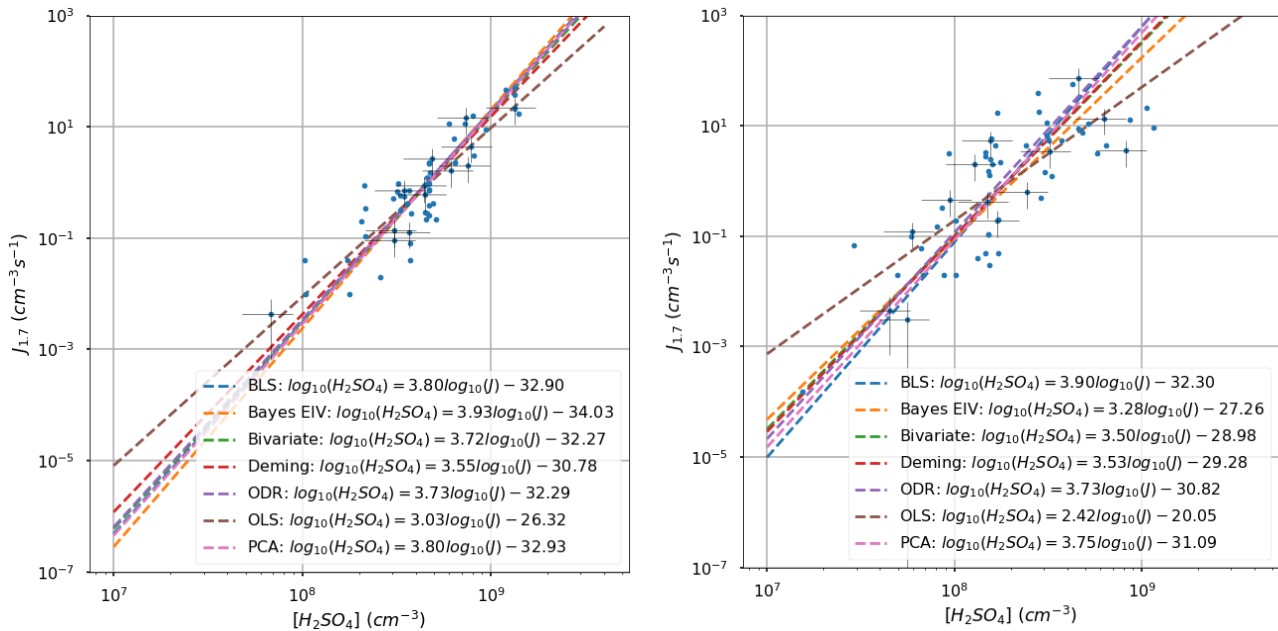

**Figure 5. Regression lines fitted to data from Dunne et al. (2016) similarly as in Fig 1. In left panel the observations are from 292K temperature and in right panel from 278K temperature.**

**Table 1. The uncertainties used in simulation for sensitivity test for increasing uncertainty**

| dataset | $\sigma_{abs}$ | $\sigma_{rel}$ | Ratio (= ($\sigma_{rel}$ * $x'_{obs}$) / $\sigma_{abs}$ ) |
|---------|----------------|----------------|-----------------------------------------------------------|
| 1 | $10^3$ | 0.05 | 315.0 |
| 2 | $10^4$ | 0.18 | 113.4 |
| 3 | $7*10^4$ | 0.3 | 27.0 |
| 4 | $4*10^5$ | 0.3 | 4.7 |
| 5 | $6.5*10^5$ | 0.45 | 4.4 |
| 6 | $10^6$ | 0.55 | 3.5 |

## Appendix A1 – Minimizing criteria for regression methods applied in the manuscript.

In this appendix, we introduce the minimizing criteria ($C_{method}$) for all methods applied in the main text. We also give the equations for regression coefficients ($\hat{\alpha}_{method}$ and $\hat{\beta}_{method}$) for the methods.

### Ordinary Least Squares (OLS)

**OLS** minimizes the sum of squares vertical distances (residuals) between each point and the fitted line. OLS regression minimizes the following criterion:

$$C_{OLS} = \sum_{i=1}^{N}\left(y_i - \hat{\alpha}_{OLS} - \hat{\beta}_{OLS}x_i\right)^2 \tag{2}$$

where $\hat{\alpha}_{OLS}$ and $\hat{\beta}_{OLS}$ refer to estimators calculated from the data, given by

$$\hat{\beta}_{OLS} = \frac{S_x}{S_y}, \hat{\alpha}_{OLS} = \bar{x} - \hat{\beta}_{OLS}\bar{y} \tag{3}$$

where observed variances for $x$ $S_x = \sum_{i=1}^{N}(x_i - \bar{x})^2$ and for $y$ $S_y = \sum_{i=1}^{N}(y_i - \bar{y})^2$, and observed covariance for $x$ and $y$ $S_{xy} =$

15 $\sum_{i=1}^{N}(x_i - \bar{x})(y_i - \bar{y})$

### Orthogonal regression (ODR)

**ODR** ([https://docs.scipy.org/doc/external/odrpack_guide.pdf](https://docs.scipy.org/doc/external/odrpack_guide.pdf), [https://docs.scipy.org/doc/scipy/reference/odr.html](https://docs.scipy.org/doc/scipy/reference/odr.html), accessed 2018-07-27) minimizes the sum of the square of orthogonal distances between each point and the line, the criteria is given by

$$C_{ODR} = \sum_{i=1}^{N}\left(\left(x_i - \frac{y_i + x_i/\hat{\beta}_{ODR} - \hat{\alpha}_{ODR}}{\hat{\beta}_{ODR} + 1/\hat{\beta}_{ODR}}\right)^2 + \left(y_i - \hat{\alpha}_{ODR} - \frac{\hat{\beta}_{ODR}y_i + x_i - \hat{\alpha}_{ODR}\hat{\beta}_{ODR}}{\hat{\beta}_{ODR} + 1/\hat{\beta}_{ODR}}\right)^2\right) \tag{4}$$

Where

$$\hat{\beta}_{ODR} = \frac{S_y - S_x + \sqrt{\left(S_y - S_x\right)^2 + 4S_{xy}^2}}{2S_{xy}} \tag{5}$$

25 and

$$\hat{\alpha}_{ODR} = \bar{y} - \hat{\beta}_{ODR}\bar{x} \tag{6}$$

ODR takes into account that errors exist in both axes but not the exact values of the variances of variables. Thus only the ratio between the two error variances ($\lambda_{xy}$) is needed to improve the methodology. With notation of Francq and Govaerts (2014)
30 this ratio is given by,

$$\lambda_{xy} = \frac{\sigma_y^2}{\sigma_x^2} \qquad (7)$$

where the numerator of the ratio is the error variance in the data in Y-axis and the denominator is the error variance in the data in X-axis.

## Deming Regression (DR)

**DR** is the ML (Maximum Likelihood) solution of Eq. 1 when $\lambda_{xy}$ is known. In practice, $\lambda_{xy}$ is unknown and it is estimated from the variances of $x$ and $y$ calculated from the data.

The DR minimizes the criterion $C_{DR}$ the sum of the square of (weighted) oblique distances between each point to the line

$$C_{DR} = \sum_{i=1}^{N} \left( \lambda_{xy} \left( x_i - \frac{y_i + \lambda_{xy}x_i/\hat{\beta}_{DR} - \hat{\alpha}_{DR}}{\hat{\beta}_{DR} + \lambda_{xy}/\hat{\beta}_{DR}} \right)^2 + \left( y_i - \hat{\alpha}_{DR} - \frac{\hat{\beta}_{DR}y_i + \lambda_{xy}x_i - \hat{\alpha}_{DR}\hat{\beta}_{DR}}{\hat{\beta}_{DR} + \lambda_{xy}/\hat{\beta}_{DR}} \right)^2 \right) \qquad (8)$$

where

$$\hat{\beta}_{DR} = \frac{S_y - \lambda_{xy}S_x + \sqrt{(S_y - \lambda_{xy}S_x)^2 + 4\lambda_{xy}S_{xy}^2}}{2S_{xy}} \qquad (9)$$

 and

$$\hat{\alpha}_{DR} = \bar{y} - \hat{\beta}_{DR}\bar{x} \qquad (10)$$

## Bivariate Least Square regression, BLS

**BLS** is a generic name but here we refer to the formulation described in Francq and Govaerts (2014) and references therein.
 BLS takes into account errors and heteroscedasticity in both axes and is written usually in matrix notation. BLS minimizes the criterion $C_{BLS}$, the sum of weighted residuals $W_{BLS}$ given by:

$$C_{BLS} = \frac{1}{W_{BLS}} \sum_{i=1}^{N} (y_i - \hat{\alpha}_{BLS} - \hat{\beta}_{BLS}x_i)^2 \qquad (11)$$

with

$$W_{BLS} = \sigma_\varepsilon^2 = \frac{\sigma_y^2}{n_y} + \hat{\beta}_{BLS}^2 \frac{\sigma_x^2}{n_x} \qquad (12)$$

Estimators for the parameters are computed by iterations with the following formulas:

$$\frac{1}{W_{BLS}} \begin{pmatrix} N & \sum_{i=1}^{N} x_i \\ \sum_{i=1}^{N} x_i & \sum_{i=1}^{N} x_i^2 \end{pmatrix} \begin{pmatrix} \hat{\alpha}_{BLS} \\ \hat{\beta}_{BLS} \end{pmatrix} = \frac{1}{W_{BLS}} \left( \sum_{i=1}^{N} \left( x_i y_i + \hat{\beta}_{BLS} \frac{\sigma_x^2}{n_x} \frac{\sum_{i=1}^{N}(y_i - \hat{\alpha}_{BLS} - \hat{\beta}_{BLS}x_i)^2}{W_{BLS}} \right) \right) \qquad (13)$$

Where known uncertainties $\sigma_x^2$ and $\sigma_y^2$ are in this study replaced with estimated variances $S_x$ and $S_y$.

A second Bivariate regression method that was used in this study is an implementation of the regression method described by **York** *et al.* (2004, Section III). The minimisation criterion is described in York *et al.* (1968) (York, D., 1968, 'Least squares fitting of a straight line with correlated errors', Earth and Planetary Science Letters (1969), pp. 320-324, North-Holland Publishing Company, Amsterdam.):

$$C_{york} = \sum_{i=0}^{N} \frac{1}{1-r_i^2}\left\{w(x_i)\left(x_{i,adj} - x_i\right)^2 - 2r\sqrt{w(x_i)w(y_i)}\left(x_{i,adj} - x_i\right)\left(y_{i,adj} - y_i\right) + w(y_i)\left(y_{i,adj} - y_i\right)^2\right\} \qquad (14)$$

Where $w(x_i) = 1/\sigma_x^2$ and $w = (y_i)1/\sigma_y^2$ are the weight coefficients for x and y, respectively, and r is the correlation coefficient between x and y. $x_{i,adj}$ and $y_{i,adj}$ are adjusted values of $x_i$, $y_i$, that fulfill the requirement

$$y_{i,adj} = \hat{\alpha}_{york} + \hat{\beta}_{york}x_{i,adj} \qquad (15)$$

The solution for $\hat{\alpha}_{york}$ and $\hat{\beta}_{york}$ is found iteratively following the ten step algorithm presented in **York** *et al.* (2004, Section III).

**The Principal Component Analysis based regression (PCA)**

PCA can be applied for bivariate and multivariate cases.

For one independent and one dependent variable, the regression line is

$y = \hat{\alpha}_{PCA} + \hat{\beta}_{PCA}x$ where the error between the *observed* value $y_i$ and *estimated* value $a+bx_i$ is minimum. For *n* points data, we compute *a* and *b* by using the method of least squares that minimizes:

$$C_{PCA} = \sum_{i=1}^{N}\left(y_i - \hat{\alpha}_{PCA} - \hat{\beta}_{PCA}x_i\right)^2 \qquad (16)$$

This is a standard technique that gives regression coefficients $\alpha$ and $\beta$.

$$\begin{bmatrix} \hat{\alpha}_{PCA} \\ \hat{\beta}_{PCA} \end{bmatrix} = \frac{\begin{bmatrix} S_x & -\bar{x} \\ -\bar{x} & 1 \end{bmatrix}}{S_x - \bar{x}^2}\begin{bmatrix} \bar{y} \\ S_{xy} \end{bmatrix} \qquad (17)$$

**Bayesian error-in-variables regression (Bayes EIV)**

**Bayes EIV** regression estimate applies Bayesian inference using the popular Stan software tool (http://mc-stan.org/users/documentation/, accessed 2018-07-27), which allowed the use of prior information of the model parameters. We assumed

$\beta_{BEIV} \sim student\_t(5, 0.0, 100.0)$

$\alpha_{BEIV} \sim student\_t(5, 0.0, 100.0)$

$x_{true} \sim lognormal(\mu_x, \sigma_x)$

$y_{true} = 10.0^{(\alpha_{BEIV} + \beta_{BEIV} * log_{10}(x_{true}))};$

where $\mu$, and $\sigma$ are the mean and standard deviation of $x_{true}$ and $y_{true}$ and are treated as unknowns. The observations $x_{obs}$ and $y_{obs}$ of $x_{true}$ and $y_{true}$, respectively, were defined as:

$x_{obs} \sim normal(x_{true}, \sigma_{rel,x}*x_{true} + \sigma_{abs,x});$

10    $y_{obs} \sim normal(y_{true}, \sigma_{rel,y}*y_{true} + \sigma_{abs,y});$

where $\sigma_{rel}$ and $\sigma_{abs}$ are the relative and absolute components of standard uncertainties, respectively.

The Stan tool solved regression problems using 1000 iterations, and it provided a posteriori distributions for the model

15    parameters $\beta_{BEIV}$ and $\alpha_{BEIV}$. For the definitions of given student_t, lognormal, and normal probability distributions, see Stan documentation. In our regression analysis, we used the maximum a posteriori estimates for $\beta_{BEIV}$ and $\alpha_{BEIV}$ provided by the software tool.