# Peer review of "Technical note: Effects of Uncertainties and Number of Data points on Line Fitting - a Case Study on New Particle Formation"

_Atmospheric Chemistry and Physics, 2018_

## Referee Comment (RC1) · Anonymous Referee #2 · 29 Dec 2018

"Technical note: Effects of Uncertainties and Number of Data points on Inference from Data – a Case Study on New Particle Formation" by Mikkonen et al. tests seven methods of linear regression on synthesized data. The resulting estimates of slope are compared among the methods and among various settings on uncertainty, sample volume and pre-screening.

This is a very nice study. It provides a concise reminder of how consequential the choice of linear regression method is. The example raised in the manuscript is easy to follow. While the subject is far from new, the atmospheric science has not seen a study with so many regression methods tested, as far as I know. The conclusions apply to a

broad range of scientific analysis, in atmospheric science and beyond. I recommend publication. I only have minor suggestions to improve the readability.

Make the title more specific by including linear regression or line fitting, in addition to, or in place of, inference.

Shorten the second sentence in Introduction and the first sentence in Conclusions.

Treat "data" as either plural or singular, but not both, in the second paragraph of Section 2.1.

Drop the "s" in "comes" in line 5, page 4.

Replace "on" with "in" in line 2, page 6.

Avoid placing the legend over the lines in Figure 3.

---

## Referee Comment (RC2) · Anonymous Referee #1 · 29 Jan 2019

**Review of "Technical Note: Effects of Uncertainties and Number of Data points on Inference from Data – a Case Study on New Particle Formation" by Mikkonen et al.**

This paper begins by discussion of using regression to infer aspects of observed data and goes on to describe the issues related to new particle formation rates. The heart of the paper involves generation of data purported to represent the logarithmic relationship between new particle formation rates and sulphuric acid concentrations. Several datasets are produced varying the amount of uncertainty, the sample size, and the number of outliers using seven regression procedures. From the regression fits, the paper makes recommendations as to when various procedures are appropriate.

This reviewer found the paper interesting and relevant for studies of atmospheric measurements, but in some cases the detail was not enough to assess the value of the results or the recommendations presented. The paper needs significant work before it is ready for publication. The authors should review the recommendations of the reviewers and make the needed changes. Perhaps the revised paper will be suitable for publication.

General comments.

One of the key points of the paper was the inclusion of accurate estimates of errors in linear regression. This reviewer found the discussion of errors significantly lacking, and indeed including some incorrect statements. Within a measurement, there are two types of error: random and systematic. Random errors can come from natural atmospheric fluctuations and instrument noise. Systematic errors can come from errors in calibration and loss of analyte in the inlet. This reviewer has never heard the term (nor could I find reference to) "natural error". One of the papers referenced (Carroll and Ruppert, 1996) also discusses "equation error", which refers to the errors associated with using an inappropriate form of a fitting equation. The paper needs a much more thorough description of errors, including introducing the symbols used later in the paper to describe errors.

The paper states on page 3, line 12 that the data used in this study are new particle formation rates and sulphuric acid concentrations. In fact, the data are simply calculations of two variables related by a linear relationship with noise added to represent random and systematic uncertainties (as done in other previous papers on linear regression). The data could represent any relation that is expected to be linear. The paper does not address nor answer any of the issues related to measurement or calculation of new particle formation rates except to say that one needs proper error estimates to perform regression on observed data, and that there are significant differences found depending on how data is handled. The reviewer finds this attempt to connect a linear regression paper to new particle formation without actually directly addressing the issue misleading. One solution would be to change the title, eliminating the part of about new particle formation, and to simply present new particle formation as one example of where error estimates are important for linear regression. With the current title, the paper needs much more emphasis on the issues related to determining new particle formation using measurements and regression procedures.

Several regression methods are used in the analysis, but the information about their use is superficial. For example, many of the methods are iterative. If proper convergence criteria are not set, then the results obtained are not useful. It is important to state the convergence criteria for each iterative method and state how it was determined that convergence was reached. For other methods, if

there are adjustable parameters, these should also be discussed. Also, the software or program used for each of the methods should be given. If they are programs written in-house, it might be appropriate to make them available to the reader.

Specific Comments
It should be mentioned, perhaps in the introduction, that linear regression is appropriate when there are two measures of the same quantity (for example, by two different instruments) or when there are two measures that are related by a physical law (for example, the dependence of the logarithm of a rate coefficient on inverse temperature).

Page 1, line 20. Suggest changing "comes" to "come" since strictly speaking "data" is plural (although often used singular).

Page 1, line 22. Did not understand the "making inferences in some more general context than was directly studied". Suggest rewording or adding more information.

Page 1, line 23. Suggest "…the bias in the analysis method…". Sentence needs period.

Page 1, line 29. After "…coefficients are underestimated…" suggest adding a reference.

Page 1, line 29-30. Suggest "Measurement error needs to be taken into account, particularly when errors are large." Suggest removing "Thus, we chose such parameters as our test variables in this study." Suggest replacing it with "To demonstrate this point, we show the effects of large errors on linear regression in this study."

Page 2, line 1. Suggest "…known to strongly affect the formation…".

Page 2, line 3. Suggest "...between $J$ and $H_2SO_4$ is typically assumed to be of the form: …".

Page 2, line 6. Suggest "…formation on global aerosol amounts and characteristics. Theoretically in homogeneous nucleation, the slope of this relationship is related to the number of sulphuric acid molecules in the nucleating critical cluster, based on the…".

Page 2, line 9. Suggest "…results have shown discrepancies in the expected $J$ vs. $H_2SO_4$ dependence."

Page 2, line 9-11. Suggest "Analysing data from Hyytiälä in 2003, Kuang et al. (2008) used an unconstrained least squares method and obtained $\beta$=1.99 for the slope, wheras Sihto et al. (2006) reported a value of 1.16 using OLS from the same field campaign."

Page 2, line 12. Suggest "…different time windows, but a significant proportion of this…".

Page 2, line 14. Suggest "…fitting method as presented in York…"

Page 2, line 15-16. Suggest "…of the methods that do not need to know the errors in advance, but instead made use of estimated variances."

Page 2, line 16. Suggest "Here, we present appropriate tools for using that approach."

Page 2, line 17. Suggest "…have been made to present methods accounting for errors in predictor variables for regression-type analysis, going back to Deming (1943)."

Page 2, line 19. Suggest "…due to its simplicity and common availability in frequently used software."

Page 2, line 20. Suggest "…methodological papers utilizing similar…".

Page 2, line 21. Suggest "…raised the awareness of the problem in the remote sensing…".

Page 2, line 22. Suggest "…follows their approach and introduces…".

Page 2, line 24. Suggest a different word that methods as it was used at the beginning of the sentence.

Page 2, line 25. Suggest "…in each variable must be taken into account using approaches called errors-in-variables (EIV) regression."

Page 2, line 30. Suggest remove "described.

Page 2, line 31. Suggest "ORDPACK is a somewhat…".

Page 2, line 32. "Mahalanobis distance" is not a term most are familiar with. Might be worth a sentence and/or a reference to explain why it is different. Alternatively, perhaps leave out that detail.

Page 3, Lines 4-25. In discussing new particle formation rates and the relationship to sulphuric acid concentrations, the authors might consider discussion the following subjects:

Are the errors in measurement of $J$ and $H_2SO_4$ related?

What is known about other factors that might affect the relationship between $J$ and $H_2SO_4$ (such as water vapor, temperature, pressure, etc.)?

Page 3, Lines 4-11. See earlier comments about errors.

Page 3, line 12. Suggest "…particle formation rates at 1.7…".

Page 3, line 13. Suggest "…concentrations simulated…".

Page 3, line 13. Suggest "…pure sulphuric acid in nucleation experiments from the CLOUD…".

Page 3, line 14. Suggest "…with corresponding expected values, their variances, and the covariance structures."

Page 3, line 15-16. It is clear you are proud of the accomplishments using CLOUD, but this reviewer suggests removing the sentence that begins "The chamber data at CERN…". Then, add CERN after "The" in the next sentence.

Page 3, line 18. The word precise is used twice in this sentence, but it does not say how precise. Given the earlier comments this reviewer made about the lack of direct connection between this study and NPF studies, perhaps the details of CERN and NPF studies could be reduced or eliminated (lines 15-20). In this discussion, the connection between $J_{1.7}$ and $H_2SO_4$ concentration is not clearly demonstrated. Is it not true that the calculation involves corrections for condensation and (for some sizes) wall loss? Suggest being more complete or leaving out this part.

Page 3, line 19. If this sentence remains in the paper, need another word or more discussion of what is meant by "inference".

Page 3, line 13. Change : to ' after $\beta$.

Page 4, line 12. Suggest "In measured data, the variables…"

Page 4, line 13. Suggest "…the measurements, and the true…." and "Thus, we use simulated data…"

Page 4, line 15. Suggest "…formation rates ($J_{1.7}$) and sulphuric acid concentrations…".

Page 4, line 20-21 and line 26. Suggest adding units to (molecules-cm$^{-3}$) to numbers.

Page 4, line 30. Suggest "This represents the quality…".

Page 4. Before starting the Results section, suggest some discussion of the fit methods, perhaps in the supplement. Suggest adding some basic introduction to the fit methods in the paper. This reviewer suggests testing the application of all the methods by testing with a known data set, such as Pearson's data with York's weights (York, 1966) whose fit parameters are known with very high accuracy.

Page 5, line 8. It is not correct to say these methods had "equal accuracy" without stating the level of accuracy, in other words plus or minus an absolute level or plus or minus a percentage.

Page 5, line 11. From the errors given in Table 1, show how the totals errors used in Figure 2 were calculated.

Page 5, line 11. Suggest "…and with varying absolute and…".

Page 5, line 14. Suggest "…significantly as more uncertainty…".

Page 5, line 16. Suggest "…quite robust with increasing…".

Page 5, line 17. Suggest "…of methods to decreasing number…".

Page 5, line 20. Suggest "…estimated slopes can be very high."

Page 5, line 20. Suggest "…slopes stabilize close to their characteristic levels (within xx% for five methods) for large datasets."

Page 5, line 21. Suggest "…more than 100 observations."

Page 5, line 22. It should be recognized that the number of points needed for a good fit depends on the uncertainties used. A few points will work fine if the uncertainties are small, while many more points are needed if uncertainties are large. This can perhaps be expressed at $\sigma_x/x$. Also, ensuring convergence is important for some of the methods (discussed above). To get an accurate representation of the data, it is also helpful for the data to cover a wide range. The x-data in this study only cover the range from about 5 to 7 ($\log_{10}[H_2SO_4]$). It would be interesting for fits when the values covered a factor of 5 to 10, even if they are not realistic for actual atmospheric situations.

Page 5, line 24. This reviewer was not sure what is meant by "high and low numbers" and "high number" in this sentence. This needs more discussion and clarity for the reader to understand clearly what was done.

Page 5, line 30. Suggest "…were not affected in either case…".

Page 5, line 31. Suggest "We did not explore how large a number of outliers would be needed to seriously disrupt the fits for the various methods. We felt that it is likely not realistic to have situations with more than 10% outliers.

Page 6, lines 2-4. This sentence needs rewording including improvement of the English to make it clear.

Page 6, line 4. Suggest "…of method should be based on the properties…".

Page 6, lines 5-8. This should be reworked based on suggestions made above.

Page 6, line 11-12. It states that the fits are made with "real" data. This is not true. These are all synthetic data. It also says that four of the methods gave slopes close to the true value. Suggest a quantitative comparison: slopes are within 5% of the true value (or whatever is appropriate). The methods are listed as good here are different than those listed in the Results section. Suggest making this consistent.

Page 6, line 14. It states that fits with small observations with all methods are highly uncertain. This does not agree with the earlier discussion and what is shown in Figure 3. Again, suggest quantitative comparisons and then statements about agreement (or lack of) that are also quantitative in this sentence and next few.

Page 6, line 15. Suggest "BLS was the most accurate…".

Page 6, line 16. Statement does not agree with the that made in Results.

Page 6, line 18. Suggest "…number of outliers (Figure 4), ODR and the York bivariate methods were the most stable…"

Page 6, line 20. Suggest "…sensitive to outliers after OLS."

Page 6, line 22. The recommendations depend on the level of uncertainty. Suggest being more quantitative, in other words, something like "When errors ($\sigma_x/x$) are greater than 50%, then method x and y performed systematically better than methods w and z."

Page 6, line 24. Suggest rewording "…we recommend considering twice…".

Page 6, line 25. Suggest "…robust with small numbers of data points." (Is this is what is meant?)

Page 6, line 32. Suggest "…were responsible for investigation…".

---

## Author Comment (AC1) · 22 Mar 2019

We thank the reviewer on the positive feedback and helpful comments. Corrections to improve the readability of the revised manuscript are made as suggested and the title is changed in more specific form: "Technical note: Effects of Uncertainties and Number of Data points on Line Fitting - a Case Study on New Particle Formation"

---

## Author Comment (AC2) · 22 Mar 2019

**Response to Reviewer 1**

We thank the reviewer for the helpful comments and suggestions. Below we address each of the reviewer's comments and indicate the requested changes to the manuscript. The reviewer's comments are marked in italic font and our response and changes to the manuscript are in plain text.

**Review of "Technical Note: Effects of Uncertainties and Number of Data points on Inference from Data – a Case Study on New Particle Formation" by Mikkonen et al.**

This paper begins by discussion of using regression to infer aspects of observed data and goes on to describe the issues related to new particle formation rates. The heart of the paper involves generation of data purported to represent the logarithmic relationship between new particle formation rates and sulphuric acid concentrations. Several datasets are produced varying the amount of uncertainty, the sample size, and the number of outliers using seven regression procedures. From the regression fits, the paper makes recommendations as to when various procedures are appropriate.

This reviewer found the paper interesting and relevant for studies of atmospheric measurements, but in some cases the detail was not enough to assess the value of the results or the recommendations presented. The paper needs significant work before it is ready for publication. The authors should review the recommendations of the reviewers and make the needed changes. Perhaps the revised paper will be suitable for publication.

**General comments.**

One of the key points of the paper was the inclusion of accurate estimates of errors in linear regression. This reviewer found the discussion of errors significantly lacking, and indeed including some incorrect statements. Within a measurement, there are two types of error: random and systematic.

Random errors can come from natural atmospheric fluctuations and instrument noise. Systematic errors can come from errors in calibration and loss of analyte in the inlet. This reviewer has never heard the term (nor could I find reference to) "natural error". One of the papers referenced (Carroll and Ruppert, 1996) also discusses "equation error", which refers to the errors associated with using an inappropriate form of a fitting equation. The paper needs a much more thorough description of errors, including introducing the symbols used later in the paper to describe errors.

A: We agree with the reviewer that the terminology with the errors needs to be clarified, as there are differences within substance areas. In statistical literature, systematic error is usually referred as "bias" and random error is divided in two parts, as reviewer indicated, which are caused by natural, stochastic, variation and the measurement itself (typically instrument or its user). The terminology is clarified in the revised manuscript, the section 2.1 now starts with text:

"Measurement data contains different types of errors. Usually, the errors are divided to two main class: random and systematic error. Systematic errors, commonly referred as bias, in experimental observations usually come from the measuring instruments. They may occur because there is something wrong with the instrument or its data handling system, or because the instrument is not used correctly by the operator. In line fitting, bias cannot be taken account but the random error may have different components, of which two are discussed here: natural error and measurement error. In addition, one should note the existence of equation error, discussed in Carroll and Ruppert (1996), which refers to using an inappropriate form of a fitting equation. Measurement error is more generally understood, it is where measured values do not fully represent the true values of variable being measured. This also contains sampling error, e.g. in the case of H2SO4 measurement the sampled air in the measurement instrument is not representative sample of outside air. Natural error is that the true connection between the two variables is has stochastic variation by some natural or physical cause e.g. certain amount of H2SO4 does not cause same number of new particles formed."

The paper states on page 3, line 12 that the data used in this study are new particle formation rates and sulphuric acid concentrations. In fact, the data are simply calculations of two variables related by a linear relationship with noise added to represent random and systematic uncertainties (as done in other previous papers on linear regression). The data could represent any relation that is expected to be linear. The paper does not address nor answer any of the issues related to measurement or calculation of new particle formation rates except to say that one needs proper error estimates to perform regression on observed data, and that there are significant differences found depending on how data is handled. The reviewer finds this attempt to connect a linear regression paper to new particle formation without actually directly addressing the issue misleading. One solution would be to change the title, eliminating the part of about new particle formation, and to simply present new particle formation as one example of where error estimates are important for linear regression. With the current title, the paper needs much more emphasis on the issues related to determining new particle formation using measurements and regression procedures.

A: The reviewer is correct that the sentence on page 3, line 12 in the original manuscript may give wrong impression on the data used, even though it stated that the data are "...concentrations simulated to mimic observations of..." and thus not claiming they are measured. We changed the beginning of the paragraph to form: "The data used in this study consist of simulated new particle formation rates at 1.7 nanometre size  $(J_{1.7})$  and sulphuric acid  $(H_2SO_4)$  concentrations mimicking observations of pure sulphuric acid in nucleation experiments..."

New particle formation data was chosen as the basis of our simulated data because we think that with these kind of data inadequate analysis methods are often used regardless of the fact that the variables contain significant uncertainties. We agree with the reviewer that the data could be any set of numbers assumed to have linear relationship but to raise the awareness in the community we need to relate the simulations to well-known datatype. We added sentence on this to the end of chapter 2.2: "However, it should be kept in mind that the data could be any set of numbers assumed to have linear relationship but to raise the awareness in the research community we related the simulations to well-known datatype." However, in this type of short technical comment we do not want to take the attention from the analysis methods to specifics in NPF formation rate calculations, which are discussed in multiple papers and textbooks.

Several regression methods are used in the analysis, but the information about their use is superficial. For example, many of the methods are iterative. If proper convergence criteria are not set, then the results obtained are not useful. It is important to state the convergence criteria for each iterative method and state how it was determined that convergence was reached. For other methods, if there are adjustable parameters, these should also be discussed. Also, the software or program used for each of the methods should be given. If

they are programs written in-house, it might be appropriate to make them available to the reader.

A: Information about which methods are iterative can be found in revised Supplement containing already the minimizing criteria. References on the software used is added to revised manuscript section 2.2: "The analysis for OLS and PCA were calculated with R-functions "Im" and "prcomp", respectively (R Core Team, 2018) DR was calculated with package deming (Therneau, 2018) and BLS with package BivRegBLS (Francq and Berger, 2017) in R. The ODR based estimates were obtained using scipy.odr python package (Jones et al. 2001-), while the python package pystan (Stan Development Team, 2018) was used for calculating the Bayesian regression estimates. Finally, the York bivariate estimates were produced with a custom python implementation of the algorithm presented by York et al. (2004). " Convergence criteria in factory built functions are kept as default set by the writers of the software.

**Specific Comments**

It should be mentioned, perhaps in the introduction, that linear regression is appropriate when there are two measures of the same quantity (for example, by two different instruments) or when there are two measures that are related by a physical law (for example, the dependence of the logarithm of a rate coefficient on inverse temperature).

A: The reviewer is correct that this is important information but we want to believe that the readers of ACP are acquaint with basics of regression/line fitting.

Page 1, line 20. Suggest changing "comes" to "come" since strictly speaking "data" is plural (although often used singular).

A: Corrected as suggested

Page 1, line 22. Did not understand the "making inferences in some more general context than was directly studied". Suggest rewording or adding more information.

A: "inferences" changed to "deductions"

Page 1, line 23. Suggest "...the bias in the analysis method...". Sentence needs period.

Page 1, line 29. After "...coefficients are underestimated..." suggest adding a reference. Page 1, line 29-30. Suggest "Measurement error needs to be taken into account, particularly when errors are large." Suggest removing "Thus, we chose such parameters as our test variables in this study." Suggest replacing it with "To demonstrate this point, we show the effects of large errors on linear regression in this study."

Page 2, line 1. Suggest "...known to strongly affect the formation ... ".

Page 2, line 3. Suggest "...between J and H2SO4 is typically assumed to be of the form: ...". Page 2, line 6. Suggest "...formation on global aerosol amounts and characteristics. Theoretically in homogeneous nucleation, the slope of this relationship is related to the number of sulphuric acid molecules in the nucleating critical cluster, based on the...".

Page 2, line 9. Suggest "...results have shown discrepancies in the expected J vs. H2SO4 dependence."

Page 2, line 9-11. Suggest "Analysing data from Hyytiälä in 2003, Kuang et al. (2008) used an unconstrained least squares method and obtained  $\beta$ =1.99 for the slope, wheras Sihto et al. (2006) reported a value of 1.16 using OLS from the same field campaign."

Page 2, line 12. Suggest "...different time windows, but a significant proportion of this...". Page 2, line 14. Suggest "...fitting method as presented in York..."

Page 2, line 15-16. Suggest "...of the methods that do not need to know the errors in advance, but instead made use of estimated variances."

Page 2, line 16. Suggest "Here, we present appropriate tools for using that approach."

Page 2, line 17. Suggest "...have been made to present methods accounting for errors in predictor variables for regression-type analysis, going back to Deming (1943)."

Page 2, line 19. Suggest "...due to its simplicity and common availability in frequently used software."

Page 2, line 20. Suggest "...methodological papers utilizing similar...".

Page 2, line 21. Suggest "...raised the awareness of the problem in the remote sensing...".

Page 2, line 22. Suggest "...follows their approach and introduces...".

Page 2, line 24. Suggest a different word that methods as it was used at the beginning of the sentence.

Page 2, line 25. Suggest "...in each variable must be taken into account using approaches called errorsin-variables (EIV) regression."

Page 2, line 30. Suggest remove "described.

A: All suggestions above are applied to the manuscript.

Page 2, line 31. Suggest "ORDPACK is a somewhat...".

Page 2, line 32. "Mahalanobis distance" is not a term most are familiar with. Might be worth a sentence and/or a reference to explain why it is different. Alternatively, perhaps leave out that detail.

A: Sentence corrected and comment on Mahalanobis distance removed

Page 3, Lines 4-25. In discussing new particle formation rates and the relationship to sulphuric acid concentrations, the authors might consider discussion the following subjects:

Are the errors in measurement of J and H2SO4 related?

*What is known about other factors that might affect the relationship between J and* H2SO4 *(such as water vapor, temperature, pressure, etc.)?*

A: In the simulated data the errors are not correlated but in the real measurements they might be. Even though the measurements are made with separate instruments, independent on each other, there might be come confounding factor effecting both of them at the same time. The factors listed by the reviewer are some of those. We added references to papers discussing these to the revised manuscript. Additionally, sentences referring to correlated error situation is added to this section: "In this study, we assume that the errors of the different variables are uncorrelated, but in some cases it has to be taken into account, as noted e.g. in Trefall and Nordö (1959) and Mandel (1984). The correlation between the errors of two variables, measured with separate instruments, independent on each other, like formation rate and H2SO4, may come e.g. from environmental variables affecting both of them at the same time. Factors affecting formation of sulphuric acid have been studied in various papers, e.g. in Weber et al. (1997) and Mikkonen et al. (2011). New particle formation rates, in turn, have been studied e.g. in Boy et al.( 2008) and in Hamed et al. (2011) and similarities between affecting factors can be seen. In addition, factors like room temperature in measurement space and atmospheric pressure may affect to measurement instruments, thus causing additional error."

Page 3, Lines 4-11. See earlier comments about errors.

A: See comment above and corresponding modifications.

Page 3, line 12. Suggest "...particle formation rates at 1.7...".
Page 3, line 13. Suggest "...concentrations simulated...".
Page 3, line 13. Suggest "...pure sulphuric acid in nucleation experiments from the CLOUD..."

A: Corrections made as suggested for comments above

Page 3, line 14. Suggest "...with corresponding expected values, their variances, and the covariance structures."

A: Corrected as suggested

Page 3, line 15-16. It is clear you are proud of the accomplishments using CLOUD, but this reviewer suggests removing the sentence that begins "The chamber data at CERN...". Then,

add CERN after "The" in the next sentence.

A: The data mimicking results from CLOUD was not used because we are proud of them but because we are concerned that many analyses on the data are made with methods not taking account the measurement errors. We added sentence on this to the end of section 2.1: "Additionally, many of the published papers on this topic do not describe how they are taking account the uncertainties in the analysis, which leaves a doubt that they are not treated properly."

Page 3, line 18. The word precise is used twice in this sentence, but it does not say how precise. Given the earlier comments this reviewer made about the lack of direct connection between this study and NPF studies, perhaps the details of CERN and NPF studies could be reduced or eliminated (lines 15-20). In this discussion, the connection between J1.7 and H2SO4 concentration is not clearly demonstrated. Is it not true that the calculation involves corrections for condensation and (for some sizes) wall loss? Suggest being more complete or leaving out this part.

Page 3, line 19. If this sentence remains in the paper, need another word or more discussion of what is meant by "inference".

A: We added explanations to the use of the term "precise": "The core is a large (volume 26m3) electro-polished stainless steel chamber with temperature control (temperature stability better than 0.1 K) at any tropospheric temperature, precise delivery of selected gases (SO2, O3, NH3, various organic compounds) and ultrapure humidified synthetic air, and very low gas-phase contaminant levels."

The connection between  $J_{1.7}$  and  $H_2SO_4$  is one of the key questions studied at CLOUD, and these studies utilize regression analyses. We chose to base our simulated datasets of  $J_{1.7}$  and  $H_2SO_4$  on data from CLOUD, because their well-controlled experiments make it possible to exclude other error sources than uncertainties on the  $J_{1.7}$  and  $H_2SO_4$ . We added clarifications on the modified manuscript: "... and  $J_{1.7}$  thus refers to the formation rate of particles as the instrument detects them, taking into account the known particle losses due to coagulation and deposition on the chamber walls. These variables were chosen because they are both known to have considerable measurement errors and their relationship is studied frequently using regression-based analyses"

Page 3, line 30. Change : to 'after  $\beta$ .

Page 4, line 12. Suggest "In measured data, the variables..."

Page 4, line 13. Suggest "...the measurements, and the true...." and "Thus, we use simulated data..."

Page 4, line 15. Suggest "...formation rates (J1.7) and sulphuric acid concentrations...". Page 4, line 20-21 and line 26. Suggest adding units to (molecules-cm-3) to numbers. Page 4, line 30. Suggest "This represents the quality...".

A: Corrections made as suggested for comments above

Page 4. Before starting the Results section, suggest some discussion of the fit methods, perhaps in the supplement. Suggest adding some basic introduction to the fit methods in the paper. This reviewer suggests testing the application of all the methods by testing with a known data set, such as Pearson's data with York's weights (York, 1966) whose fit

parameters are known with very high accuracy.

A: We extended the descriptions on the methods to section 2.2 in the revised manuscript, as indicated in answers above. The reason for using simulated dataset in this study was that we would know exactly the expected value for slope and the errors. Thus using Pearson's data would not give that much additional value for the manuscript.

Page 5, line 8. It is not correct to say these methods had "equal accuracy" without stating the level of accuracy, in other words plus or minus an absolute level or plus or minus a percentage.

A: The sentence corrected to form: "The best performing methods with equal accuracy, i.e. within 2% range, were ODR ( $\beta_{ODR}$ =3.27), Bayes EIV ( $\beta_{BEIV}$ =3.24) and BLS ( $\beta_{BLS}$ =3.22), whereas York ( $\beta_{York}$ =3.15) was within 5% range, but Deming ( $\beta_{DR}$ =2.95) and PCA ( $\beta_{PCA}$ =2.92) slightly underestimated the slope."

Page 5, line 11. From the errors given in Table 1, show how the totals errors used in Figure 2 were calculated.

A: The relative errors (Fig. 2 horizontal axis values) were actually calculated from the simulated dataset values (as mentioned in Fig. 2 label) with  $\frac{|x_{obs} - x_{true}|}{x_{true}}$  and not directly from the absolute and relative uncertainty values given in Table 1. That is, first each of the simulated data sets were generated as described in Section 3 and then the relative errors were calculated from the data itself.

Page 5, line 11. Suggest "...and with varying absolute and...".
Page 5, line 14. Suggest "...significantly as more uncertainty...".
Page 5, line 16. Suggest "...quite robust with increasing...".
Page 5, line 17. Suggest "...of methods to decreasing number...".
Page 5, line 20. Suggest "...estimated slopes can be very high."
Page 5, line 20. Suggest "...slopes stabilize close to their characteristic levels (within xx% for five methods) for large datasets."

A: Corrections made as suggested for comments above

Page 5, line 21. Suggest "...more than 100 observations."

A: Corrected as suggested

Page 5, line 22. It should be recognized that the number of points needed for a good fit depends on the uncertainties used. A few points will work fine if the uncertainties are small, while many more points are needed if uncertainties are large. This can perhaps be expressed at  $\sigma_x/x$ . Also, ensuring convergence is important for some of the methods (discussed above).

A: A sentence on the relationship of number of observations and the uncertainties was added at the end of this paragraph in the revised manuscript: "Though, it should be remembered that number of points needed for a good fit depends on the uncertainties in the data." Discussion on convergence was already added to method descriptions.

To get an accurate representation of the data, it is also helpful for the data to cover a wide range. The xdata in this study only cover the range from about 5 to 7 (log10[H2SO4]). It would be interesting for fits when the values covered a factor of 5 to 10, even if they are not realistic for actual atmospheric situations.

A: Wider range for X-data does not change the phenomenon, neither does varying the expected value of the "true slope". These were tested when preparing the manuscript and thus we are showing only atmospherically relevant numbers.

Page 5, line 24. This reviewer was not sure what is meant by "high and low numbers" and "

high number " in this sentence. This needs more discussion and clarity for the reader to understand clearly what was done.

A: Unclear terms replaced by "high or low end of the distribution" and "large number" in the revised manuscript

Page 5, line 30. Suggest "...were not affected in either case...".

A: Corrected as suggested

Page 5, line 31. Suggest "We did not explore how large a number of outliers would be needed to seriously disrupt the fits for the various methods. We felt that it is likely not realistic to have situations with more than 10% outliers.

A: Corrected as suggested

Page 6, lines 2-4. This sentence needs rewording including improvement of the English to make it clear.

A: Sentence written in form: "Ordinary least squares regression can be used to answer some simple questions on data, such as is *Y* related to *X* but if we are interested on the strength of the

relationship and the predictor variable *X* contains some error, then error-in-variables methods should be applied"

Page 6, line 4. Suggest "...of method should be based on the properties...".

A: Corrected as suggested

Page 6, lines 5-8. This should be reworked based on suggestions made above.

A: Definition of term "natural error" was inserted to the text as written above.

Page 6, line 11-12. It states that the fits are made with "real" data. This is not true. These are all synthetic data. It also says that four of the methods gave slopes close to the true value. Suggest a quantitative comparison: slopes are within 5% of the true value (or whatever is appropriate).

The methods are listed as good here are different than those listed in the Results section. Suggest making this consistent.

A: The sentence means that the errors for the simulated data are taken from real measurements. The sentence is reformulated in the revised manuscript in order to avoid confusion, quantitative comparison is inserted as suggested and consistency with Results section is ensured. The new sentence is: "In Fig 1, we showed that in case of simulations where errors are taken from real measurements of J1.7 and H2SO4 four of the methods gave slopes within 5% of the "true" known value: BLS, York bivariate, Bayes EIV and ODR."

Page 6, line 14. It states that fits with small observations with all methods are highly uncertain. This does not agree with the earlier discussion and what is shown in Figure 3. Again, suggest quantitative comparisons and then statements about agreement (or lack of) that are also quantitative in this sentence and next few.

A: Uncertainty ranges of all methods in Fig. 3 are relatively high with small numbers of observations, even though average performance of BLS and, at some extent, York method are close to "true slope"

Page 6, line 15. Suggest "BLS was the most accurate...".

A: Corrected as suggested

Page 6, line 16. Statement does not agree with the that made in Results.

A: Statement in Results section was amended to be consistent with the conclusion

Page 6, line 18. Suggest "...number of outliers (Figure 4), ODR and the York bivariate methods were the most stable..."

A: Corrected as suggested

Page 6, line 20. Suggest "...sensitive to outliers after OLS."

A: Corrected as suggested

Page 6, line 22. The recommendations depend on the level of uncertainty. Suggest being more quantitative, in other words, something like "When errors ( $\sigma_x/x$ ) are greater than 50%,

then method x and y performed systematically better than methods w and z."

A: Recommendations quantified in the revised manuscript into form: "If the errors are not known, and they are estimated from data, BLS and ODR showed out to be the most robust in cases of increasing uncertainty (relative error rE > 30% in Fig 2) and with high number of outliers. In our test data, BLS and ODR stayed stable up to rE >80% in Fig. 2 whereas DR and PCA started to be more uncertain when rE > 30% and Bayes EIV when rE>50%. "

Page 6, line 24. Suggest rewording "...we recommend considering twice...".

A: removed word "twice"

Page 6, line 25. Suggest "...robust with small numbers of data points." (Is this is what is meant?)
Page 6, line 32. Suggest "...were responsible for investigation...".

A: Corrections made as suggested for both comments

---

## Referee Report (RR1)

**Review of "Technical Note: Effects of Uncertainties and Number of Data Points on Line Fitting – a Case Study on New Particle Formation", revised, by Mikkonen et al.**

This is a review of a revised paper comparing various linear regression methods that account for errors in the x- and y-variables, which are also compared with ordinary least squares (OLS). The paper introduces the problem of linear regression with errors in both variables, and describes the method they used for generating synthetic data that is meant to represent data collected in new particle formation (NPF) studies (similar means, distributions, and noise levels). The results of the various fits are compared using various sized data sets, various noise levels, and data with extreme outliers included. Conclusions are drawn and recommendations made as to the preferred regression method(s) to use in particular situations encountered with NPF data.

**General Comments.**

This is a revised paper, changed according to comments made by two reviewers. The recommendations of the reviewers were acknowledged, but not all the suggested changes were implemented. This reviewer has no problem with this, but clear and justifiable reasons for not making recommended changes should be stated. This was not always the case in the author's response to the reviewers. This is discussed below.

In making the changes that were suggested, sometimes the English was not carefully checked. The original version of the paper had some minor issues with English. The revised version has many more problems. This reviewer suggests a careful review of the English, perhaps with the help of a native English speaker.

The heart of the paper depends on the analysis of synthetic data and the methods for its generation and simulation of noise. It is clearly explained what was done, but it is not always clear why. This reviewer suggests a detailed re-write of section 3 to make clear the decisions made on the approaches used to generate the data. Also, information should be added to the supplement to show the impacts of the data generation process on the final data. My concerns in this regard are addressed below. Also included below are other issues with the revised paper.

In general, the paper as written is quite short. It can easily be expanded to include additional important information that would make it more useful to researchers using various linear regression methods. This reviewer recommends that the authors feel free to provide any important information that would be useful to potential readers of their paper.

**Synthetic Data Generation**

The generation of synthetic data is routinely used to test data analysis methods. This is a suitable approach and can be useful in uncovering errors. If the data are not produced properly, bias and errors can result even when the analysis approach is correct. Consider the simple case of a series of measurements of a quantity near the detection limit. The calculation of the $H_2SO_4$ concentrations as described on page 5 is a good example. One thousand data points are randomly selected from a log-normal distribution with mean of $2 \times 10^6$ molecules-$cm^{-3}$. The standard deviation is $2.4 \times 10^6$. This generates concentrations from the mid-$10^4$ to the low-$10^7$ range. The recovered mean ranges from about 1.9 to $2.1 \times 10^6$, and the standard deviation from about 2.0 to $2.8 \times 10^6$. These values show a range because of the finite size of the data set and the random selection of data from the distribution. Next noise is added that has two components: a constant factor (representing the baseline noise) and a factor proportional to the value (representing signal-carried noise). The noise is selected from normal distributions with means of $4.0 \times 10^5$ for the constant part and 0.3 times the value for the proportional part. The noise ranges from about $-10^7$ to $+10^7$. Note that the errors are about the size of the largest values generated from the log-normal distribution. This is not realistic, since data below the detection limit are typically filtered out, and data considered valid are well above the baseline noise (typically 3 standard deviations). Indeed, there are several points in Figure 1 of the paper for

$H_2SO_4$ values less than $10^5$ cm$^{-3}$. Note that the $J_{1.7}$ values do not follow the trend of the fit lines for these low $H_2SO_4$ values. This is because there are nucleation rates calculated that are negative and are eliminated from the log-log plot that would balance these values. In any case, this generates data whose lower values are negative (about 40-60 values out of 1000) and are undefined in logarithmic space. This creates a small bias with the mean of non-negative data about 3% to 10% larger than that used to generate the data. The standard deviation is 1% to 15% larger. The effect is even greater for the calculated nucleation rates (about 270 to 300 negative values) because of the 3.3 multiplier and the negative intercept. The mean for the nucleation rate is about 1% to 70% larger than the noise-free data set, while the standard deviation is 0.1 to 1.9 times that in the noise-free data. These biases affect the quality of the linear fits.

This reviewer suggests a different procedure for generating synthetic data. Rather than sample from a normal distribution, suggest generating evenly spaced data from some minimum value above the detection limit to some typical largest value observed. This would produce data that evenly covers the range of expected values, rather than a clumping data near the mean value of a distribution. In looking at the paper by Kurten (ACP, 2019), it appears that room temperature $H_2SO_4$ concentrations in CLOUD NPF experiments range from 3 x $10^7$ to $10^9$ cm$^{-3}$. Based on the noise levels assigned in this paper (Mikkonen et al.), this appears to be reasonable with a range from the detection limit (about 3 times the noise of small values) to the largest value measured. In tests, this reviewer configured the data (1000 points) evenly spaced in the logarithmic domain, with natural logarithms ranging from 16 to 21. This produced a mean $H_2SO_4$ concentration of about 2.6 x $10^8$ cm$^{-3}$ and a standard deviation of about 3.2 x $10^8$ cm$^{-3}$. This results in 0 to 1 values that are negative out of 1000. Calculations of the nucleation rate only produces about 20 to 30 negative numbers. This leads to data that are much more reasonable (see Figure R1) in that the data are scattered evenly without unbalanced "tails", particularly at low $H_2SO_4$ values. It is worth noting that the nucleation rates are very large at $10^9$ cm$^{-3}$ $H_2SO_4$, but these are the values obtained from the equation $\log_{10}(J_{1.7}) = 3.3*\log_{10}(H_2SO_4)-23$ provided in the paper.

[Figure]

Figure R1. Synthetic data generated evenly spaced in logarithm space. Includes noise added to $H_2SO_4$ and $J_{1.7}$ values as described in Mikkonen et al., revised. OLS fit yields a slope of about 3.1, an intercept of about -21, and $r^2$ of about 0.9.

Note that least squares routines have problems if the dynamic range is not large compared to the noise in the data. In the above example, if the data range is significant narrowed, the data show no obvious trend, and OLS gives a distinctly flatter slope (Figure R2).

[Figure]

Figure R2. Synthetic data generated as in Figure R1, but with minimum and maximum $\ln(H_2SO_4)$ =15 and 16, respectively. OLS fit yields slopes of about 1.2, intercepts of about -9, and $r^2$ of 0.2. This appears to be caused by data covering a narrow dynamic range compared to the noise level.

In the generation of synthetic data, there is a random factor associated with the noise estimation (selection from normal distributions). This means that there is variability to the data and thus to the fits. It would be beneficial to run several (perhaps tens or even hundreds) 1000 data-point sets and give information on the variability of the fit parameters (minimum, maximum, mean, standard deviation of the slopes and intercepts of the various methods). This would allow other researchers to have better understanding of the range of values that can be expected for the different methods.

Since the authors likely have access to significant NPF data from CLOUD and Hyytiala, it is possible they could perform fits using what they believe to be the appropriate fitting method and compare them to the literature values. This would call attention to using the correct procedure, and would provide a database of corrected data for use by the aerosol community. This reviewer does not believe this is beyond the scope of this paper. It would involve a paragraph of introduction to the data, a description of the fit method selected, and a table of results.

**Weighting**

One of the advantages of many of the non-OLS fitting methods is the ability to weight data based on some factor, typically the inverse of the uncertainty. This can minimize the effects of outliers and cause the fits to depend more on data that is more certain. This reviewer suggests a discussion of weighting be included, and its impact on fitting be demonstrated, including comparisons of fits with and without weighting, and the impact of different weighting approaches on the fit results.

**Addressing comments from original paper**

This reviewer felt that several comments from the original paper were not properly addressed. This is not acceptable. The role of the reviewer is to make sure the presentation is scientifically robust and the paper justifies the approaches taken. It is important the sufficient information be given that the research can be reproduced. There were several comments that this reviewer does not believe were adequately addressed in the author's response to the reviews, and in changes to the paper.

1. Comment about more information on iterative methods.
   The response indicated that the information requested was in the Supplement. This is mostly not the case. While the functions to be minimized are given for many of the methods, the

criteria for convergence are not indicated. Also, no information is given on the York method, but only reference to the York, 2004 paper. Iterative methods rely on convergence criteria that indicate if sequential iterations vary by less than some fractional value, the procedure is halted. If incorrect convergence criteria are used, the procedure could be halted prematurely They also are prescribed a maximum number of iterations and initial guesses for the parameters. It is possible that if the maximum number is reached (because too small a value was selected) before convergence is reached, then the fit values will be incorrect. Poorly selected initial values can also inhibit convergence. This reviewer suggests using a large maximum number of iterations and repeating the fits with convergence criteria that are gradually tightened to see if the fits are changed substantially. In any case, the paper and/or the supplement need to address clearly the issue of using iterative methods, and indicate the convergence criteria and number of iterations prescribed for each one. This reviewer does not feel that keeping the convergence criteria at the defaults by the software programmers is prudent, sensible, or sufficient to address the concerns. Sensitivity tests must be done. Also, the statement about the York method (a custom python implementation) in particular calls for tests with known data sets to ensure it is functioning as it should. More detail about the York method also needs to be added to the Supplement section.

        The equation given for deriving the fit parameters for the BLS (Francq and Govaerts, 2014) does not agree precisely with their Equation 24. Suggest checking Lisy et al., 1990 and other related papers to make sure equation is correct.

2. In the author's comments (page 8) related to the original manuscript page 4, it is stated that the "true" values are known because of the synthetic generation procedure used. While this is true before the noise was added to each variable, it is not necessarily true afterwards. This is because of the issues discussed above in which negative data are eliminated when conversion to logarithm space, which can potentially create biases. This is the argument for using other data in which the slope and intercept are known and established. This reviewers insists that at least one other data set be tested with each of the methods and compared with the known, exact fit parameters.

**Other comments.**

        Suggest defining terms that might not be familiar to atmospheric scientists, such as "homoscedasticity" and "heteroscedasticity". The terms "estimators" and "predictors" are also used without definition, as is "a posteriori".

Manuscript specific comments
Page 1, line 10. Suggest eliminating "on a scatterplot" to read "Fitting a line of two measured…"
Page 1, line 10. Suggest removing "as", removing "considered" and changing "simplest" to "most common" to read "…variables is one of the most common statistical…"
Page 1, line 21. Suggest "Atmospheric measurements always come with some measurement error."
Page 1, line 23. Suggest rewording and/or adding text to clarify what is meant by "ill-formulated".
Page 1, line 25. Suggest "Regression models can be linear or non-linear the selection of which depends on the data being analyzed."
Page 1, lines 26-27. Suggest "…that the independent variable of the model has been measured without error and the model accounts only for…"
Page 1, line 29. It is not clear why OLS should be asymptotic, since it is not an iterative method. OLS can be close to the correct parameter value if the noise in the independent variable is small. Suggest rewording this sentence.
Page 2, line 1. Suggest changing the sentence "Measurement error needs to be taken into account" to something that indicates that methods that account for measurement error in the independent variable need to be utilized. In the next sentence that starts "Thus, we chose such…", suggest changing

to something that says that test data were developed that included significant uncertainties in both the independent and dependent variables.

Page 2, line 4. Suggest removing "as" to read "…is typically assume to be…"

Page 2, line 5. Suggest giving the units for J and $H_2SO_4$.

Page 2, line 7. Suggest "…to estimate the effects of new particle…"

Page 2, line 11. Suggest adding information to describe what is meant by "unconstrained" in this context.

Page 2, lines 13-14. Suggest "…of this inconsistency is very likely due to use of different fitting methods." Also, "…has been acknowledged previously in…"

Page 2, line 26. EIV methods simply mean that errors in both variables are accounted for. Suggest a statement that says this.

Page 2, line 33. This sentence implies that ORDPACK is unique in accounting for point by point variance and covariance, but other methods also have the capability (including York). Suggest rewording.

Page 3, line 7. It is misleading to say that in linear regression bias cannot be taken into account. Indeed, it is important to minimize bias through careful and regular calibrations and zeros. Analysis of these data can reveal information about baseline noise levels and signal carried noise. At a minimum, an upper limit to the amount of bias can be estimated, although obviously not known with absolute certainty. Suggest rewording.

Page 3, line 11. "…of the variable being measured."

Page 3, line 13. Suggest rewording "Natural error is that the true connection…" to something like "Natural error is the variability caused by natural or physical phenomenon"

Page 3, line 16. Suggest "…when interpreting fits."

Page 3, lines 17-18. Suggest "…in some cases this has to be taken…"

Page 3, line 19. Suggest "…independent of each other…"

Page 3, line 23. Suggest "…room temperature in the measurement space and atmospheric pressure may affect the performance of instrumentation…"

Page 3, lines 28-29. In my previous review, the point was that the sentence about the CERN NPF data is not necessary and should be eliminated.

Page 4, lines 1-2. Suggest rewording the sentence starting with "Existing data…" to something like "The existing data on NPF includes what are believed to be the most important routes that involve sulfuric acid, ammonia and water vapor…"

Page 4, line 6. Rather than "These variables…" suggest "The relationships between precursor gas-phase concentrations and particle formation rates were chosen for study because…". Suggest eliminating "…which makes them good illustrative variables for this study."

Page 4, lines 9-10. Suggest rewording to something like "…in the analyses, which casts doubt that errors have been treated properly."

Page 4, line 11. Suggest "…to have a linear relationship, but in order to raise awareness in the aerosol research community, in this study we relate our analysis to the important problem of understanding new particle formation."

Page 4, lines 24-25. Suggest "…how they account for measurements errors. The minimizing criteria for all methods are given in supplement S1, but here we give the basic principles of the methods."

Page 4, line 28. Suggest "…the error variances, $\lambda_{xy}$, of the variables…"

Page 4, line 29. Suggest "The approach of PCA is…"

Page 4, line 33. The phrase "linear scale uncertainties in logarithmic scale regression" needs explanation. Suggest adding a line or two to clarify, and also to explain why the York method cannot account for this.

Page 5, line 3. Suggest "…in both variables, and thus is a more advanced method than DR…". Note than many of these methods allow entering point by point uncertainties and thus can handle heteroscedasticity.

Page 5, lines 4-5. This sentence doesn't make sense. One method accounts for measurement variance, while the other methods require estimates of measurement errors. Isn't this the same thing. Suggest rewording.

Page 5, line 5. Suggest "Though for Bayes…"

Page 5, line 6. Suggest removing comma "…be applied with both errors given…"

Page 5, line 8. Suggest "…was calculated with the package…and BLS with the package…". Consider putting the software package names in quotes or underlining to make it clear they are special words.

Page 5, line 14. Upper case for the independent and dependent variables, but that is different than equation 1 and the supplement. Suggest making consistent throughout.

Page 5, line 24. When this reviewer performed statistics on the noise free values calculated for $J_{1.7}$, there was considerable variability from one 1000 point data set to the next. Giving the mean and standard deviation for one data set is rather meaningless. Either perform statistics on many data sets and given means of the individual means, or eliminate this sentence.

Page 5, line 26. The word "true" is used here to mean the simulated measured values (data with errors included) whereas elsewhere "true" is used to mean the input values (without errors). Suggest a different word than "true" here, and use care to be consistent throughout the paper.

Page 5, line 28. The second sigma should be changed to "$\sigma_{rel,y}$".

Page 5, line 32. Outliers are also generated on the low tail of the distribution. These should be eliminated as well.

Page 6, lines 4-6. In a list, suggest using "first…second…third" or "firstly…secondly…thirdly". In other words, be consistent.

Page 6, line 8. Suggest "noise-free slope" instead "true slope" for reasons mentioned above.

Page 6, lines 14-15. Suggest "…and performed fits with each method on all of these datasets."

Page 6, lines 15-16. Suggest "…to Figure 1 marked in black."

Page 6, lines 16-17. Suggest rewording the sentence that begins "It shows that when…" since the first half and the second half say the same thing, just reversed.

Page 6, line 17. Suggest "…which overestimates the slope…"

Page 6, line 19. Suggest "…are not changing significantly."

Page 6, lines 23-24. Suggest "…to their characteristic levels…"

Page 6, line 25. This recommendation is misleading because it depends on the noise (error) level of the data. For the conditions of this study, it may be true, but could be very different for data with more or less noise. Suggest rewording.

Page 7, line 7. Suggest '…on data, such as "How is Y related to X?".'

Page 7, lines 9-10. Suggest rewording or eliminating "…because methods measure slightly different things about the data." It is not clear what is meant by this sentence.

Page 7, line 9. The conclusion that including of error in the analysis will never lead to a more biased estimator was not discussed in the paper, nor is it justified by the material presented. Suggest either adding a discussion of this point with data that proves it, or eliminate the sentence.

Page 7, lines 20-21. It is not true that all fit methods highly uncertain with small numbers of points. Some methods are exact with small numbers of points (e.g. York method and Pearson's data with York's weights). This needs to be reworded or eliminated. Also, true of related statements in lines 31-32.

Page 7, line 29. A new symbol was introduced without definition "rE". Suggest either eliminating or defining. This could have been defined and used earlier in the paper.

Page 7, line 32. Suggest "Our study showed BLS to be the most robust with…". Maybe true, but depends on the uncertainty of the data.

Page 8, lines 1-2. It is not clear how error distributions are used in the Bayes EIV method. Indeed, there are symbols used in the supplement describing the Bayes method that are not defined. This sentence and the related section in the supplement need to be reworded.

Supplement specific comments.

Throughout the supplement, define variables used. Also, equations need to be numbered so they can be referred to.

Page 1, line 3. Suggest and introductory paragraph that describes what is to follow for each of the methods. Suggest headers to separate the discussion of the different methods. Suggest some more discussion of the synthetic data generation such as showing probability density functions for each variable graphically and perhaps other useful information that would be helpful to the reader.

Page 1, line 15. The sentence that starts "ODR takes into account…" does not make sense. This needs rewording to make clear. How can the errors be accounted for but not the variances?

Page 1, line 18. There is no error in the Y-axis, only error in the Y-data. Suggest making this change here and throughout the paper.

Page 2, BLS section. The first equation looks exactly like OLS with weights. I'm sure there is more to it than that, so more explanation is needed. As stated before, the second equation doesn't agree exactly with that in the Francq and Govaerts, 2014 paper.

Page 2, line 9. Suggest "A second bivariate regression method that was used in this study is an…"

Page 2, PCA section. This again looks just like OLS. Suggest adding more text to explain how it is different.

Page 2, line 26. Suggest "…and are treated as unknowns."

Page 3, line 1. Suggest "The Stan tool solved regression problems using 1000 iterations, and it provided…". Also suggest "In our analyses, we used the maximum a posteriori estimates for $\beta$ and $\beta$ provided by the software tool." What is the iterative formula used in this method? How do you know 1000 iterations is enough for full convergence? Is there a convergence criterion that indicates the software can finish? How do you know the criterion is appropriate? (These questions are related to general comments made earlier, but they apply to each of these methods.)

---

## Author Response (AR2)

Review of "Technical Note: Effects of Uncertainties and Number of Data Points on Line Fitting – a Case Study on New Particle Formation", revised, by Mikkonen et al.

This is a review of a revised paper comparing various linear regression methods that account for errors in the x- and y-variables, which are also compared with ordinary least squares (OLS). The paper introduces the problem of linear regression with errors in both variables, and describes the method they used for generating synthetic data that is meant to represent data collected in new particle formation (NPF) studies (similar means, distributions, and noise levels). The results of the various fits are compared using various sized data sets, various noise levels, and data with extreme outliers included. Conclusions are drawn and recommendations made as to the preferred regression method(s) to use in particular situations encountered with NPF data.

A: We thank the reviewer for the helpful comments and suggestions. Our answers to the concerns addressed are below. The questions/comments from the reviewer are in italic font and our answers follow with plain text.

**General Comments.**

This is a revised paper, changed according to comments made by two reviewers. The recommendations of the reviewers were acknowledged, but not all the suggested changes were implemented. This reviewer has no problem with this, but clear and justifiable reasons for not making recommended changes should be stated. This was not always the case in the author's response to the reviewers. This is discussed below.

In making the changes that were suggested, sometimes the English was not carefully checked. The original version of the paper had some minor issues with English. The revised version has many more problems. This reviewer suggests a careful review of the English, perhaps with the help of a native English speaker.

A: In this response we will elaborate more the reasons why all suggested changes were not made. The English is also checked more carefully.

The heart of the paper depends on the analysis of synthetic data and the methods for its generation and simulation of noise. It is clearly explained what was done, but it is not always clear why. This reviewer suggests a detailed re-write of section 3 to make clear the decisions made on the approaches used to generate the data. Also, information should be added to the supplement to show the impacts of the data generation process on the final data. My concerns in this regard are addressed below. Also included below are other issues with the revised paper.

In general, the paper as written is quite short. It can easily be expanded to include additional important information that would make it more useful to researchers using various linear regression methods. This reviewer recommends that the authors feel free to provide any important information that would be useful to potential readers of their paper.

A: In our opinion, technical notes are supposed to be short, concise pieces of information, which can be then applied for use in wider studies. We have added important information to text and supplement, with the help of comments of the reviewer, but we do not want to lengthen the manuscript too much.

**Synthetic Data Generation**

The generation of synthetic data is routinely used to test data analysis methods. This is a suitable approach and can be useful in uncovering errors. If the data are not produced properly, bias and errors can result even when the analysis approach is correct. Consider the simple case of a series of measurements of a quantity near the detection limit. The calculation of the  $H_2SO_4$  concentrations as described on page 5 is a good example. One thousand data points are randomly selected from a lognormal distribution with mean of 2 x  $10^6$  molecules-cm-3. The standard deviation is 2.4 x  $10^6$ . This generates concentrations from the mid- $10^4$  to the low- $10^7$  range. The recovered mean ranges from about 1.9 to 2.1 x  $10^6$ , and the standard deviation from about 2.0 to 2.8 x  $10^6$ . These values show a range because of the finite size of the data set and the random selection of data from the distribution. Next noise is added that has two components: a constant factor (representing the baseline noise) and a factor proportional to the value (representing signal-carried noise). The noise is selected from normal distributions with means of 4.0 x 105 for the constant part and 0.3 times the value for the proportional part. The noise ranges from about  $-10^7$  to  $+10^7$ . Note that the errors are about the size of the largest values generated from the lognormal distribution. This is not realistic, since data below the detection limit are typically filtered out, and data considered valid are well above the baseline noise (typically 3 standard deviations). Indeed, there are several points in Figure 1 of the paper for  $H_2SO_4$  values less than  $10^5$  cm-3. Note that the  $J_{1.7}$ values do not follow the trend of the fit lines for these low  $H_2SO_4$  values. This is because there are nucleation rates calculated that are negative and are eliminated from the log-log plot that would balance these values. In any case, this generates data whose lower values are negative (about 40-60 values out of 1000) and are undefined in logarithmic space. This creates a small bias with the mean of non-negative data about 3% to 10% larger than that used to generate the data. The standard deviation is 1% to 15% larger. The effect is even greater for the calculated nucleation rates (about 270 to 300 negative values) because of the 3.3 multiplier and the negative intercept. The mean for the nucleation rate is about 1% to 70% larger than the noise-free data set, while the standard deviation is 0.1 to 1.9 times that in the noisefree data. These biases affect the quality of the linear fits.

A: The selected simulation method resulted in distributions that are observed in atmosphere, e.g. for  $H_2SO_4$  the distributions were similar as in our previous study (Mikkonen et al. ACP, 2011, 11, 11319-11334. doi:10.5194/acp-11-11319-2011). When simulating the data, negative values were immediately replaced by new simulated values until only non-negative values existed, which indeed offsets the simulated data from perfectly symmetric distributions in Fig 1. However, perfect symmetry is typically not a requirement for real datasets, hence we believe it should not be the case for our simulated data set either. It is obvious, that the shape of the data distributions will affect the performance of various regression estimators, but studying all possible related nuances would be a very laborious task. After all, the goal of this paper is to provide a simple demonstration of a frequently disregarded statistical phenomenon, not to exhaust all its possible variations that readers may encounter.

This reviewer suggests a different procedure for generating synthetic data. Rather than sample from a normal distribution, suggest generating evenly spaced data from some minimum value above the detection limit to some typical largest value observed. This would produce data that evenly covers the range of expected values, rather than a clumping data near the mean value of a distribution. In looking at the paper by Kurten (ACP, 2019), it appears that room temperature  $H_2SO_4$  concentrations in CLOUD NPF experiments range from 3 x 107 to 109 cm-3. Based on the noise levels assigned in this paper (Mikkonen et al.), this appears to be reasonable with a range from the detection limit (about 3 times the noise of small values) to the largest value measured. In tests, this reviewer configured the data (1000 points)

evenly spaced in the logarithmic domain, with natural logarithms ranging from 16 to 21. This produced a mean  $H_2SO_4$  concentration of about 2.6 x  $10^8$  cm-3 and a standard deviation of about 3.2 x  $10^8$  cm-3. This results in 0 to 1 values that are negative out of 1000. Calculations of the nucleation rate only produces about 20 to 30 negative numbers. This leads to data that are much more reasonable (see Figure *R1*) in that the data are scattered evenly without unbalanced "tails", particularly at low  $H_2SO_4$  values. It is worth noting that the nucleation rates are very large at  $10^9$  cm-3  $H_2SO_4$ , but these are the values obtained from the equation  $log_{10}(J_{1.7}) = 3.3*log_{10}(H_2SO_4)-23$  provided in the paper.

Figure R1. Synthetic data generated evenly spaced in logarithm space. Includes noise added to  $H_2SO_4$  and  $J_{1.7}$  values as described in Mikkonen et al., revised. OLS fit yields a slope of about 3.1, an intercept of about -21, and  $r^2$  of about 0.9.

Note that least squares routines have problems if the dynamic range is not large compared to the noise in the data. In the above example, if the data range is significant narrowed, the data show no obvious trend, and OLS gives a distinctly flatter slope (Figure R2).

Figure R2. Synthetic data generated as in Figure R1, but with minimum and maximum  $ln(H_2SO_4) = 15$  and 16, respectively. OLS fit yields slopes of about 1.2, intercepts of about -9, and  $r^2$  of 0.2. This appears to be caused by data covering a narrow dynamic range compared to the noise level.

In the generation of synthetic data, there is a random factor associated with the noise estimation (selection from normal distributions). This means that there is variability to the data and thus to the fits. A: We thank the reviewer about the suggestion for new data simulation method. The method would assume that  $H_2SO_4$  measurement would produce uniformly distributed data, which is not the case. Due to this, and other reasons listed above, we will keep the simulation method as is.

It would be beneficial to run several (perhaps tens or even hundreds) 1000 data-point sets and give information on the variability of the fit parameters (minimum, maximum, mean, standard deviation of the slopes and intercepts of the various methods). This would allow other researchers to have better understanding of the range of values that can be expected for the different methods.

A: In fact, Fig. 3 already visualizes the information on the variability of the slope estimate with repeated random datasets. For instance, the rightmost points on the figure show the medians and the shading shows the 1 std range of slope values for 40 random subsamples with 500 values in each sample. At 500 values per sample, the variability of the slope is already quite small and doubling the sample size to 1000, as the reviewer suggests, would further narrow the variability. In our view, Fig. 3 already gives a good visual understanding of the range of slope values that can be expected for the different methods.

Since the authors likely have access to significant NPF data from CLOUD and Hyytiala, it is possible they could perform fits using what they believe to be the appropriate fitting method and compare them to the literature values. This would call attention to using the correct procedure, and would provide a database of corrected data for use by the aerosol community. This reviewer does not believe this is beyond the scope of this paper. It would involve a paragraph of introduction to the data, a description of the fit method selected, and a table of results.

A: It is true that we have access to these type of data but currently, as the data are not open, we do not have permission to use them in this study. We have made tests with data collected from Hyytiälä and San Pietro Capofiume, Italy and the results are similar than seen here with simulated data. However, we cannot publish these results and thus we will publish a Python tool for running some of the methods and encourage people to test that on their own data. The tool can be found in GitHub: https://gist.github.com/mikkopitkanen/da8c949571225e9c7093665c9803726e The link is also added to the end of the manuscript

The link is also added to the end of the manuscript.

**Weighting**

One of the advantages of many of the non-OLS fitting methods is the ability to weight data based on some factor, typically the inverse of the uncertainty. This can minimize the effects of outliers and cause the fits to depend more on data that is more certain. This reviewer suggests a discussion of weighting be included, and its impact on fitting be demonstrated, including comparisons of fits with and without weighting, and the impact of different weighting approaches on the fit results.

A: We pointed out the effect of weighting in Conclusions section and give a reference to study demonstrating effects of it with sentence: "Weighting the data based on some factor, typically the inverse

of the uncertainty, reduces the effect of outliers and makes the regression depend more on the data that is more certain (see e.g. Wu and Yu, 2018) but it does not solve the problem completely."

**Addressing comments from original paper**

This reviewer felt that several comments from the original paper were not properly addressed. This is not acceptable. The role of the reviewer is to make sure the presentation is scientifically robust and the paper justifies the approaches taken. It is important the sufficient information be given that the research can be reproduced. There were several comments that this reviewer does not believe were adequately addressed in the author's response to the reviews, and in changes to the paper.

Comment about more information on iterative methods.

The response indicated that the information requested was in the Supplement. This is mostly not the case. While the functions to be minimized are given for many of the methods, the criteria for convergence are not indicated. Also, no information is given on the York method, but only reference to the York, 2004 paper. Iterative methods rely on convergence criteria that indicate if sequential iterations vary by less than some fractional value, the procedure is halted. If incorrect convergence criteria are used, the procedure could be halted prematurely They also are prescribed a maximum number of iterations and initial guesses for the parameters. It is possible that if the maximum number is reached (because too small a value was selected) before convergence is reached, then the fit values will be incorrect. Poorly selected initial values can also inhibit convergence. This reviewer suggests using a large maximum number of iterations and repeating the fits with convergence criteria that are gradually tightened to see if the fits are changed substantially. In any case, the paper and/or the supplement need to address clearly the issue of using iterative methods, and indicate the convergence criteria and number of iterations prescribed for each one. This reviewer does not feel that keeping the convergence criteria at the defaults by the software programmers is prudent, sensible, or sufficient to address the concerns. Sensitivity tests must be done. Also, the statement about the York method (a custom python implementation) in particular calls for tests with known data sets to ensure it is functioning as it should. More detail about the York method also needs to be added to the Supplement section.

A: We agree with the reviewer that the selection of the convergence criteria is important in order to get reliable results. However, we feel that for the ready-made packages e.g. in R we can trust the developers that they have set the criteria on optimal level. In addition, we feel that every researcher conducting these kind of analyses should by default be aware of the nature of iterative methods and issues with convergence. These are taught in basic courses of data analysis. We will add description of York method in the supplement and clarify other descriptions of methods where needed. The tool for applying the methods will also be available, as stated in comment above. Also, substantial number of tests with known data were conducted with the custom York (2004) implementation to ensure that the program works as expected by using simulated data but also the so called Pearson's data set shows for example in Cantrell (2008) Fig 1. The source code already published on Github allows readers to verify the code themselves: https://gist.github.com/mikkopitkanen/da8c949571225e9c7093665c9803726e

The equation given for deriving the fit parameters for the BLS (Francq and Govaerts, 2014) does not agree precisely with their Equation 24. Suggest checking Lisy et al., 1990 and other related papers to make sure equation is correct.

A: We thank the reviewer for pointing out the typo in the equation, which is now corrected in the revised supplement.

In the author's comments (page 8) related to the original manuscript page 4, it is stated that the "true" values are known because of the synthetic generation procedure used. While this is true before the noise was added to each variable, it is not necessarily true afterwards. This is because of the issues discussed above in which negative data are eliminated when conversion to logarithm space, which can potentially create biases. This is the argument for using other data in which the slope and intercept are known and established. This reviewers insists that at least one other data set be tested with each of the methods and compared with the known, exact fit parameters.

A: In the simulation of the data, the values were generated one by one and each negative value was immediately replaced with a new simulated value, until only non-negative values were included in the data set. This is why no negative data values exist in the first place and the conversion to logarithmic scale did not eliminate any data. Still, the referee is correct, that even with this method significant number of negative values would cause bias but in our data the number was so small that the effect of negligible

**Other comments.**

Suggest defining terms that might not be familiar to atmospheric scientists, such as "homoscedasticity" and "heteroscedasticity". The terms "estimators" and "predictors" are also used without definition, as is "a posteriori".

A: Term "homoscedasticity" replaced with "equal variances". Terms "heteroscedasticity" and "a posteriori" defined in section 2.2. Terms "estimators" and "predictors" should be known by all who are conducting regression analysis.

**Manuscript specific comments**

Page 1, line 10. Suggest eliminating "on a scatterplot" to read "Fitting a line of two measured..." A: Corrected as suggested

Page 1, line 10. Suggest removing "as", removing "considered" and changing "simplest" to "most common" to read "...variables is one of the most common statistical..."

A: This would change the meaning of the sentence. It is supposed to indicate that line fitting is not that simple as people usually think.

Page 1, line 21. Suggest "Atmospheric measurements always come with some measurement error." A: Corrected as suggested

*Page 1, line 23. Suggest rewording and/or adding text to clarify what is meant by "ill-formulated".* A: Sentence changed to form: "If the relationship is not defined correctly, the inference is not valid either."

Page 1, line 25. Suggest "Regression models can be linear or non-linear the selection of which depends on the data being analyzed."

A: The sentence changed to form: "Regression models can be linear or non-linear, depending on the relationship between data sets that are analysed."

Page 1, lines 26-27. Suggest "...that the independent variable of the model has been measured without error and the model accounts only for..." A: Corrected as suggested

Page 1, line 29. It is not clear why OLS should be asymptotic, since it is not an iterative method. OLS can be close to the correct parameter value if the noise in the independent variable is small. Suggest rewording this sentence.

A: The asymptotics here refer to number of data. For clarity, the word "asymptotically" is replaced as: "with very high number of number of data points"

Page 2, line 1. Suggest changing the sentence "Measurement error needs to be taken into account" to something that indicates that methods that account for measurement error in the independent variable need to be utilized. In the next sentence that starts "Thus, we chose such…", suggest changing to something that says that test data were developed that included significant uncertainties in both the independent variables.

A: The sentences reformulated as: "If predictor variables in regression analysis contain any measurement error, methods that account for errors should be applied. Particularly when errors are large. Thus, test variables in this study were chosen such that they included significant uncertainties in both the independent and dependent variables."

Page 2, line 4. Suggest removing "as" to read "...is typically assume to be..."
Page 2, line 5. Suggest giving the units for J and H2SO4.
Page 2, line 7. Suggest "...to estimate the effects of new particle..."
A: Corrected as suggested

*Page 2, line 11. Suggest adding information to describe what is meant by "unconstrained" in this context.* A: We are not certain which unconstrained method was used, a note on this was added to text.

Page 2, lines 13-14. Suggest "...of this inconsistency is very likely due to use of different fitting methods." Also, "...has been acknowledged previously in..."

A: Corrected as suggested

- Page 2, line 26. EIV methods simply mean that errors in both variables are accounted for. Suggest a statement that says this.
- A: Added sentence: "EIV methods simply mean that errors in both variables are accounted for."
- Page 2, line 33. This sentence implies that ORDPACK is unique in accounting for point by point variance and covariance, but other methods also have the capability (including York). Suggest rewording.
- A: The sentence is comparing ORDPACK only to classical orthogonal regression. For clarity, the sentence is reworded as: "ORDPACK is a somewhat improved version of classical orthogonal regression,

so that arbitrary covariance structures are acceptable and is specifically set up so that a user can specify measurement error variances and covariance point by point, as some of the methods in this study are doing in linear analysis."

- Page 3, line 7. It is misleading to say that in linear regression bias cannot be taken into account. Indeed, it is important to minimize bias through careful and regular calibrations and zeros. Analysis of these data can reveal information about baseline noise levels and signal carried noise. At a minimum, an upper limit to the amount of bias can be estimated, although obviously not known with absolute certainty. Suggest rewording.
- A: In the actual line fitting, the bias cannot be taken account but it needs to be minimized, as the reviewer noted, with calibrations and zeros or by data pre-processing. For clarity, the sentence is reworded as: "In line fitting, bias cannot be taken account but it needs to be minimized through careful and regular instrument calibrations and zeros or data pre-processing."

Page 3, line 11. "...of the variable being measured."

- Page 3, line 13. Suggest rewording "Natural error is that the true connection..." to something like "Natural error is the variability caused by natural or physical phenomenon"
- Page 3, line 16. Suggest "...when interpreting fits."
- Page 3, lines 17-18. Suggest "...in some cases this has to be taken..."
- Page 3, line 19. Suggest "...independent of each other..."
- Page 3, line 23. Suggest "...room temperature in the measurement space and atmospheric pressure may affect the performance of instrumentation..."
- Page 3, lines 28-29. In my previous review, the point was that the sentence about the CERN NPF data is not necessary and should be eliminated.
- Page 4, lines 1-2. Suggest rewording the sentence starting with "Existing data..." to something like "The existing data on NPF includes what are believed to be the most important routes that involve sulfuric acid, ammonia and water vapor..."
- Page 4, line 6. Rather than "These variables..." suggest "The relationships between precursor gasphase concentrations and particle formation rates were chosen for study because...". Suggest eliminating "...which makes them good illustrative variables for this study."
- Page 4, lines 9-10. Suggest rewording to something like "...in the analyses, which casts doubt that errors have been treated properly."
- Page 4, line 11. Suggest "...to have a linear relationship, but in order to raise awareness in the aerosol research community, in this study we relate our analysis to the important problem of understanding new particle formation."
- Page 4, lines 24-25. Suggest "...how they account for measurements errors. The minimizing criteria for all methods are given in supplement S1, but here we give the basic principles of the methods."
- *Page 4, line 28. Suggest "…the error variances,*  $\lambda_{xy}$ *, of the variables…"*
- Page 4, line 29. Suggest "The approach of PCA is ... "

A: Corrected as suggested

Page 4, line 33. The phrase "linear scale uncertainties in logarithmic scale regression" needs explanation. Suggest adding a line or two to clarify, and also to explain why the York method cannot account for this.

A: The sentence is changed to form "However, using ODR allows for performing regression on a user defined model, while the York (2004) solution works only on linear models. This, for instance, enables using linear scale uncertainties in ODR in this study, while the York (2004) approach could only use log scale uncertainties."

- Page 5, line 3. Suggest "...in both variables, and thus is a more advanced method than DR...". Note than many of these methods allow entering point by point uncertainties and thus can handle heteroscedasticity.
- A: Corrected as suggested
- Page 5, lines 4-5. This sentence doesn't make sense. One method accounts for measurement variance, while the other methods require estimates of measurement errors. Isn't this the same thing. Suggest rewording.
- A: The sentence reworded as "PCA accounts only for the observed variance in data, whereas ODR, Bayes EIV and York bivariate regression require known estimates for measurement errors."

Page 5, line 5. Suggest "Though for Bayes..."Page 5, line 6. Suggest removing comma "...be applied with both errors given..."A: Corrected as suggested

- Page 5, line 8. Suggest "...was calculated with the package...and BLS with the package...". Consider putting the software package names in quotes or underlining to make it clear they are special words.
- A: Corrected as suggested
- Page 5, line 14. Upper case for the independent and dependent variables, but that is different than equation 1 and the supplement. Suggest making consistent throughout.

A: Corrected to lower case

- Page 5, line 24. When this reviewer performed statistics on the noise free values calculated for  $J_{1.7}$ , there was considerable variability from one 1000 point data set to the next. Giving the mean and standard deviation for one data set is rather meaningless. Either perform statistics on many data sets and given means of the individual means, or eliminate this sentence.
- A: The sentence removed as suggested
- Page 5, line 26. The word "true" is used here to mean the simulated measured values (data with errors included) whereas elsewhere "true" is used to mean the input values (without errors). Suggest a different word than "true" here, and use care to be consistent throughout the paper.
- A: "true" changed as "noise-free" as suggested in later comment

*Page 5, line 28. The second sigma should be changed to "\sigma\_{rel,y}".* A: Corrected as suggested

Page 5, line 32. Outliers are also generated on the low tail of the distribution. These should be eliminated as well.

A: This is how it was done; we clarified it to text by talking about absolute error

Page 6, lines 4-6. In a list, suggest using "first...second...third" or "firstly...secondly...thirdly". In other words, be consistent.

Page 6, line 8. Suggest "noise-free slope" instead "true slope" for reasons mentioned above.
Page 6, lines 14-15. Suggest "...and performed fits with each method on all of these datasets."
Page 6, lines 15-16. Suggest "...to Figure 1 marked in black."
A: Corrected as suggested

Page 6, lines 16-17. Suggest rewording the sentence that begins "It shows that when..." since the first half and the second half say the same thing, just reversed.

A: The sentence is changed to form: "It shows that when the uncertainty is small, the bias in OLS fit is smaller but when more uncertainty is added to data the bias increases significantly"

Page 6, line 17. Suggest "...which overestimates the slope..." Page 6, line 19. Suggest "...are not changing significantly." Page 6, lines 23-24. Suggest "...to their characteristic levels..." A: Corrected as suggested

- Page 6, line 25. This recommendation is misleading because it depends on the noise (error) level of the data. For the conditions of this study, it may be true, but could be very different for data with more or less noise. Suggest rewording.
- A: We elaborated the recommendation to cases with similar uncertainty by adding this to the end of the sentence: "...and similar type of uncertainty than presented here"

*Page 7, line 7. Suggest '…on data, such as "How is Y related to X?".* A: Corrected as suggested

Page 7, lines 9-10. Suggest rewording or eliminating "…because methods measure slightly different things about the data." It is not clear what is meant by this sentence.

A: The sentence reworded as: "...because the methods behave slightly differently with different types of error"

- Page 7, line 9. The conclusion that including of error in the analysis will never lead to a more biased estimator was not discussed in the paper, nor is it justified by the material presented. Suggest either adding a discussion of this point with data that proves it, or eliminate the sentence.
- A: The sentence is removed from the manuscript.

Page 7, lines 20-21. It is not true that all fit methods highly uncertain with small numbers of points. Some methods are exact with small numbers of points (e.g. York method and Pearson's data with York's weights). This needs to be reworded or eliminated. Also, true of related statements in lines 31-32.

A: The sentence refers to simulated data in Fig 3. It is true that for single dataset methods can be exact but even when the sample in the analysis is drawn from the same distribution containing some error, the methods will have uncertainty in the analysis. Some methods are more robust for this than others. Already Cantrell (2008) stated that for methods recommended there: "the accuracy of the slope improves with the number of data points (not so with the standard least squares with significant errors in the x-variable)" The sentence is reformulated as "The main message to learn in Fig 3 is that if the data contain some error, then with small numbers of observations all fit methods are highly uncertain."

Page 7, line 29. A new symbol was introduced without definition "rE". Suggest either eliminating or defining. This could have been defined and used earlier in the paper.

A: symbol defined as "relative error" already in the manuscript line 29. This is the first time this measure is needed.

Page 7, line 32. Suggest "Our study showed BLS to be the most robust with...". Maybe true, but depends on the uncertainty of the data.

A: The sentence changed to form "with chosen uncertainties in our simulation tests BLS showed out to be the most robust with small numbers of data points."

Page 8, lines 1-2. It is not clear how error distributions are used in the Bayes EIV method. Indeed, there are symbols used in the supplement describing the Bayes method that are not defined. This sentence and the related section in the supplement need to be reworded.

A: We reworded both the sentence in the conclusions and the supplement section regarding the Bayes EIV method and provided with additional symbol definitions.

**Supplement specific comments.**

- *Throughout the supplement, define variables used. Also, equations need to be numbered so they can be referred to.*
- A: The variables are now defined and the equations numbered.
- Page 1, line 3. Suggest and introductory paragraph that describes what is to follow for each of the methods. Suggest headers to separate the discussion of the different methods. Suggest some more discussion of the synthetic data generation such as showing probability density functions for each variable graphically and perhaps other useful information that would be helpful to the reader.

A: In the beginning, we added sentence "In this supplement, we introduce the minimizing criteria  $(C_{method})$  for all methods applied in the main text. We also give the equations for regression coefficients  $(\hat{\alpha}_{method} \text{ and } \hat{\beta}_{method})$  for the methods." In addition, headers are added for each method.

Page 1, line 15. The sentence that starts "ODR takes into account..." does not make sense. This needs rewording to make clear. How can the errors be accounted for but not the variances?

A: The sentence corrected to form: "ODR takes into account that errors exist in both axes but not the exact values of the variances of variables."

Page 1, line 18. There is no error in the Y-axis, only error in the Y-data. Suggest making this change here and throughout the paper.
A: Corrected as suggested

A: Corrected as suggested

- Page 2, BLS section. The first equation looks exactly like OLS with weights. I'm sure there is more to it than that, so more explanation is needed. As stated before, the second equation doesn't agree exactly with that in the Francq and Govaerts, 2014 paper.
- A: The equation is corrected as stated above.

Page 2, line 9. Suggest "A second bivariate regression method that was used in this study is an..." A:Corrected as suggested

Page 2, PCA section. This again looks just like OLS. Suggest adding more text to explain how it is different. A: The difference is in the coefficients  $\hat{\alpha}$  and  $\hat{\beta}$ , which are defined below.

Page 2, line 26. Suggest "...and are treated as unknowns." A:Corrected as suggested

Page 3, line 1. Suggest "The Stan tool solved regression problems using 1000 iterations, and it provided...". Also suggest "In our analyses, we used the maximum a posteriori estimates for  $\beta$  and  $\beta$  provided by the software tool." What is the iterative formula used in this method? How do you know 1000 iterations is enough for full convergence? Is there a convergence criterion that indicates the software can finish? How do you know the criterion is appropriate? (These questions are related to general comments made earlier, but they apply to each of these methods.)

A: Corrected as suggested. The convergence criteria are given in Stan documentation and, as stated above, we trust the software developers that the criteria are valid.

**Technical note: Effects of Uncertainties and Number of Data points on Line Fitting - a Case Study on New Particle Formation**

Santtu Mikkonen1, Mikko R. A. Pitkänen1,2, Tuomo Nieminen1§, Antti Lipponen2, Sini Isokääntä1, Antti Arola2, and Kari E. J. Lehtinen1,2

[revised manuscript text omitted]

---

## Author Response (AR3)

*I went through the 2nd revision of the manuscript and replies to the comments. I feel that this technical note provides some useful insights with regard to the uncertainties in line fitting of observational data, commonly used in deriving empirical relationship between new particle formation rates and precursor gas concentrations. However, I feel that several important issues raised by the previous reviewer were not adequately addressed. In particular, I think that the following issues should be addressed before the manuscript can be recommended for publication in ACP.*

**A:** We thank the reviewer for the helpful comments and suggestions. The new analysis of the Dunne et al. data made the manuscript significantly stronger. Our answers to the concerns addressed are below. The questions/comments from the reviewer are in italic font and our answers follow with plain text.

1. *Data generation, Figures R1 & R2. The previous reviewer used about two pages to support his/her arguments with regard to the importance of the data generation procedure. I consider this to be an important and valid point. In contrast, the authors' reply was too brief and not fully addressed the concerns raised. The authors should at least add a case to use the data generated according to the procedure suggested by the previous reviewer (evenly spaced rather than normal distribution). This may change the conclusion with regard to OLS, as demonstrated in Fig. R1 and Fig. R2.*

**A:** Only cases where evenly spaced data would be possible with atmospheric data is extremely well controlled experiments where concentrations could be controlled with high accuracy. In this kind of experiment the uncertainties are on a different level than with "normal" measurement data. However, we will show results for such data as the reviewer suggested in Supplement S1 and summarize them in results section with following text.

"We also applied an alternative method for simulating the data for testing different methods. The main difference compared to our method was, that the distribution of noise-free $H_2SO_4$ followed uniform distribution in log-space. With this assumption, it could be seen that OLS works almost equally well compared to EIV methods introduced here if the range of data is wide ($H_2SO_4$ concentration in range $10^6$-$10^9$). However, when scaled to concentrations usually measured in the atmosphere ($10^4$-$10^7$) the high uncertainties caused similar behaviour to data than seen in our previous simulations. Details of these results can be seen in Supplement S1."

*2. With regard to using NPF data from CLOUD and Hyytiala, as original comments and replies copied below.*
*C: Since the authors likely have access to significant NPF data from CLOUD and Hyytiala, it is possible they could perform fits using what they believe to be the appropriate fitting method and compare them to the literature values. This would call attention to using the correct procedure, and would provide a database of corrected data for use by the aerosol community. This reviewer does not believe this is beyond the scope of this paper. It would involve a paragraph of introduction to the data, a description of the fit method selected, and a table of results.*

*A: It is true that we have access to these type of data but currently, as the data are not open, we do not have permission to use them in this study. We have made tests with data collected from Hyytiälä and San Pietro Capofiume, Italy and the results are similar than seen here with*

*simulated data. However, we cannot publish these results and thus we will publish a Python tool for running some of the methods and encourage people to test that on their own data. The tool can be found in GitHub: https://gist.github.com/mikkopitkanen/da8c949571225e9c7093665c9803726e. The link is also added to the end of the manuscript.*

*I agree with the previous reviewer that it will be useful to perform fits to the real NPF data from CLOUD and Hyytiala and compare with the literature values. I feel that the authors' explanation that they do not have permission to use them is not adequate. Firstly, many papers with regard to CLOUD and Hyytialla have already been published and these data should be available for others to use based on the data policy of many journals. Secondly, some CLOUD data are in the public domain (for example, see the supplement materials of Dunne et al., Science, 2016) and can definably be used. This manuscript will be enhanced and much more useful to the community if the authors "could perform fits using what they believe to be the appropriate fitting method and compare them to the literature values".*

**A:** We made tests for the data published in Dunne et al. as the reviewer suggested. The results are implemented in the manuscript. Following text was inserted as data section 3.2:

**"3.2 Case study on measured data**

In order to show that the results gained with simulated data are applicable also in real measurement data, we applied our methods to data measured in CLOUD chamber and published by Dunne et al. (2016). Their Fig 1. shows nucleation rates ($J$) at 1.7 nm mobility diameter as a function of sulphuric acid concentration. We used their measurements with no added ammonia in two different temperatures, 278K and 292K, show in their Fig 1 subplots (D) and (E) and published as supplemental data."

In addition, following text was inserted as results section 4.2:

"Figure 5 shows the fits on the data from Dunne et al (2016). As expected, the fit with OLS is underestimated with both temperatures ($\beta_{ols}$(278K) = 2.4 and $\beta_{ols}$(292K) = 3.0). The regression equations for all methods are shown in Table/Figure. Dunne et al. did not use linear fit in their study but applied nonlinear Levenberg-Marquardt algorithm (Moré, 1978) instead on function $J_{1.7}$ = $k*[H_2SO_4]^\beta$ where $k$ is temperature dependent rate coefficient with nonlinear function including three estimable parameters (see section 8 in their Supplement for details). Thus, the results are not directly comparable as, for simplicity, we made the fits to data measured in different temperatures separately. However, their $\beta$-value for the fit ($\beta$=3.95) is quite close to our results with EIV-methods, especially slopes from Bayes EIV in 292K and BLS and PCA in both temperatures were within 5% range."

*3. First page of the reply, second line from the bottom. "We have added important information to text and supplement". The previous reviewer asked for "This reviewer suggests a detailed re-write of section 3 to make clear the decisions made on the approaches used to generate the data." I didn't find substantial revision in Section 3. Also the supplement (S1) is totally on regression method, rather than on the data generation.*

**A:** We added following text to section 3.1 after the second last paragraph:

[revised manuscript text omitted]

---

## Author Response (AR4)

*Comments to the Author:*
*Dear Authors,*

*Appreciate your efforts in addressing several important comments raised by previous reviewers and improving the manuscript. I think that the technical note is useful to the community with regard to the proper application of linear regression approaches in interpreting observational data. I have one major comment and a few minor correction/clarification.*

A: We thank the editor on the comments and the suggestions to improve the manuscript. Our response and modifications to text are shown below.

*Major comment:*
*In the conclusion section, you gave a recommendation on which method(s) should be chosen. In this regard, can you add some discussions in describing Figure 5 (Section 4.2) which method is most appropriate for the CLOUD? Is there a temperature dependence in the beita value? According to your discussion, the number of data points (observations) is important. Have you looked into the CLOUD data at lower temperature where data points are limited?*
*Also I would suggestion that you add a few sentences about the application of named methods to the CLOUD data and findings in the conclusion (section 5).*

A: We have added following text to the end of chapter 4.2 discussing on CLOUD results:

> "We also made some tests for data measured in lower temperatures, results not shown here. But as a summary, the slopes did not vary drastically from those in $\beta_{ols}$(278K) and $\beta_{ols}$(292K) when the other conditions were similar, even though the lower number of observations in lower temperatures increased uncertainty in the data. However, the intercepts $\beta_0(T)$ varied between temperatures."

Additionally, we inserted discussion, as suggested, to the end of chapter 5:

> "We also made a case study on data measured in CLOUD chamber and published by Dunne et al. (2016). In these analyses, we saw that our recommended methods above are performing best also for these data. Our tests indicated that the slope $\beta_1$ for the fit is not highly sensitive for changes in temperature in the chamber but the intercept $\beta_0$ in linear fit is. This dependency was also seen, and taken account, in Dunne et al. (2006)."

*A few minor comments:*
*(1) Page 6, Line 23: show ◊shown?*

A: Corrected as suggested

*(2) Page 8, Line 7: "in Table/Figure", which Table, which Figure? Please be specific. You meant the figure insertions?*

A: Corrected to "Figure 5"

[revised manuscript text omitted]

Where $w(x_i) = 1/\sigma_x^2$ and $w = (y_i) 1/\sigma_y^2$ are the weight coefficients for x and y, respectively, and r is the correlation coefficient between x and y. $x_{i,adj}$ and $y_{i,adj}$ are adjusted values of $x_i$, $y_i$, that fulfill the requirement

$$y_{i,adj} = \hat{\alpha}_{york} + \hat{\beta}_{york} x_{i,adj} \tag{15}$$

The solution for $\hat{\alpha}_{york}$ and $\hat{\beta}_{york}$ is found iteratively following the ten step algorithm presented in **York** *et al.* (2004, Section III).

**The Principal Component Analysis based regression (PCA)**

PCA can be applied for bivariate and multivariate cases.

For one independent and one dependent variable, the regression line is

$y = \hat{\alpha}_{PCA} + \hat{\beta}_{PCA} x$ where the error between the *observed* value $y_i$ and *estimated* value $a + bx_i$ is minimum. For $n$ points data, we compute $a$ and $b$ by using the method of least squares that minimizes:

$$C_{PCA} = \sum_{i=1}^{N}\left(y_i - \hat{\alpha}_{PCA} - \hat{\beta}_{PCA} x_i\right)^2 \tag{16}$$

This is a standard technique that gives regression coefficients $\alpha$ and $\beta$.

$$\begin{bmatrix} \hat{\alpha}_{PCA} \\ \hat{\beta}_{PCA} \end{bmatrix} = \frac{\begin{bmatrix} S_x & -\bar{x} \\ -\bar{x} & 1 \end{bmatrix}}{S_x - \bar{x}^2} \begin{bmatrix} \bar{y} \\ S_{xy} \end{bmatrix} \tag{17}$$

**Bayesian error-in-variables regression (Bayes EIV)**

**Bayes EIV** regression estimate applies Bayesian inference using the popular Stan software tool (http://mc-stan.org/users/documentation/, accessed 2018-07-27), which allowed the use of prior information of the model parameters. We assumed

$\beta_{BEIV} \sim$ student_t(5, 0.0, 100.0)
$\alpha_{BEIV} \sim$ student_t(5, 0.0, 100.0)
$x_{true} \sim$ lognormal($\mu_x$, $\sigma_x$)
$y_{true} = 10.0^{(\alpha_{BEIV} + \beta_{BEIV} * \log_{10}(x_{true}))}$;

where $\mu$, and $\sigma$ are the mean and standard deviation of $x_{true}$ and $y_{true}$ and are treated as unknowns. The observations $x_{obs}$ and $y_{obs}$ of $x_{true}$ and $y_{true}$, respectively, were defined as:

$x_{obs} \sim$ normal($x_{true}$, $\sigma_{rel,x} * x_{true} + \sigma_{abs,x}$);
$y_{obs} \sim$ normal($y_{true}$, $\sigma_{rel,y} * y_{true} + \sigma_{abs,y}$);

where $\sigma_{rel}$ and $\sigma_{abs}$ are the relative and absolute components of standard uncertainties, respectively.

The Stan tool solved regression problems using 1000 iterations, and it provided a posteriori distributions for the model parameters $\beta_{BEIV}$ and $\alpha_{BEIV}$. For the definitions of given student_t, lognormal, and normal probability distributions, see Stan documentation. In our regression analysis, we used the maximum a posteriori estimates for $\beta_{BEIV}$ and $\alpha_{BEIV}$ provided by the software tool.